# Bioactive Glial-Derived Neurotrophic Factor from a Safe Injectable Collagen–Alginate Composite Gel Rescues Retinal Photoreceptors from Retinal Degeneration in Rabbits

**DOI:** 10.3390/md22090394

**Published:** 2024-08-30

**Authors:** Tingyu Hu, Ting Zhou, Rajesh Kumar Goit, Ka Cheung Tam, Yau Kei Chan, Wai-Ching Lam, Amy Cheuk Yin Lo

**Affiliations:** 1Department of Ophthalmology, Li Ka Shing Faculty of Medicine, The University of Hong Kong, Hong Kong, China; tingyuhu@connect.hku.hk (T.H.); caroltz@connect.hku.hk (T.Z.); btkc0380@gmail.com (K.C.T.); jchanyk@hku.hk (Y.K.C.); waiching.lam@vch.ca (W.-C.L.); 2Jules Stein Eye Institute, Los Angeles, CA 90095, USA; 3Department of Ophthalmology and Visual Sciences, University of British Columbia, Vancouver, BC V5Z 3N9, Canada

**Keywords:** collagen–alginate composite hydrogel, doxycycline, electroretinography, encapsulated cell therapy, HEK cells, intravitreal, neuroprotection, retina, sustained drug delivery

## Abstract

The management of vision-threatening retinal diseases remains challenging due to the lack of an effective drug delivery system. Encapsulated cell therapy (ECT) offers a promising approach for the continuous delivery of therapeutic agents without the need for immunosuppressants. In this context, an injectable and terminable collagen–alginate composite (CAC) ECT gel, designed with a Tet-on pro-caspase-8 system, was developed as a safe intraocular drug delivery platform for the sustained release of glial-cell-line-derived neurotrophic factor (GDNF) to treat retinal degenerative diseases. This study examined the potential clinical application of the CAC ECT gel, focusing on its safety, performance, and termination through doxycycline (Dox) administration in the eyes of healthy New Zealand White rabbits, as well as its therapeutic efficacy in rabbits with sodium-iodate (SI)-induced retinal degeneration. The findings indicated that the CAC ECT gel can be safely implanted without harming the retina or lens, displaying resistance to degradation, facilitating cell attachment, and secreting bioactive GDNF. Furthermore, the GDNF levels could be modulated by the number of implants. Moreover, Dox administration was effective in terminating gel function without causing retinal damage. Notably, rabbits with retinal degeneration treated with the gels exhibited significant functional recovery in both a-wave and b-wave amplitudes and showed remarkable efficacy in reducing photoreceptor apoptosis. Given its biocompatibility, mechanical stability, controlled drug release, terminability, and therapeutic effectiveness, our CAC ECT gel presents a promising therapeutic strategy for various retinal diseases in a clinical setting, eliminating the need for immunosuppressants.

## 1. Introduction

A number of diseases affecting the posterior segment of the eye, such as age-related macular degeneration (AMD) and retinitis pigmentosa (RP), pose a chronic threat to vision and demand extended treatment periods, among which photoreceptor cells are the most affected. This presents a significant financial and labor-intensive challenge to both patients and physicians due to the extensive costs and efforts required for their treatment and management [1]. Current treatment options primarily involve administering medication through eye drops or intravitreal injections [2]. While applying eye drops is non-invasive and can be easily achieved by patients themselves, the necessity for frequent application often results in poor compliance [3,4]. Intravitreal injection is the current method for delivering drugs to the posterior segment of the eye [5]. However, to maintain effective drug levels, this invasive procedure must be performed regularly, which can lead to various cumulative risks and complications, including inflammation, endophthalmitis, retinal detachment, and cataracts [6]. As chronic retinal diseases necessitate multiple and repeated intravitreal injections for attaining effective local drug levels, there has been a growing interest in intraocular drug delivery systems [7].

Encapsulated cell therapy (ECT) is a promising strategy to provide continuous and freshly synthesized drug delivery to the targeted site by encapsulating drug-producing cells into a semipermeable membrane. The porous membrane enables oxygen and nutrients exchange to support cell survival and drug diffusion, while preventing immune cell access into the capsule without the requirement for immunosuppressants [8]. ECT, first introduced in the 1960s, has been explored for various diseases [8,9,10,11,12,13]. Despite notable progress in this field, its clinical application is still somewhat challenging to achieve. Long-term implantation of ECT systems often leads to device failure, attributed to issues with biocompatibility, system stability, cell viability, and drug production. In the realm of eye care, two ECT devices, NT-501 (Renexus^®^) and NT-503-3, developed by Neurotech Pharmaceuticals, were designed to administer ciliary neurotrophic factor (CNTF) and a soluble anti-vascular endothelial growth factor receptor, respectively [14]. These devices, which anchor to the sclera, consist of synthetic polymers (polyethylene terephthalate and polyethersulfone), along with genetically engineered human retinal pigment epithelium cells (ARPE-19) [14]. Although NT-501 has been shown to be safe and is capable of local drug release without the need for immunosuppression, it failed to achieve the primary goals in clinical trials for dry AMD and RP, leading to the discontinuation of its development for these conditions [15]. Similarly, a Phase II study for wet AMD using NT-503-3 was halted prematurely due to a high incidence of rescue interventions and insufficient patient responses [16]. While CNTF treatment may enhance photoreceptor cell survival, potential adverse effects, such as diminished electroretinogram (ERG) responses and miosis, could impact other aspects of a patient’s visual function [17]. Therefore, there is a significant demand for innovative and improved treatment options for retinal degenerative diseases (RDDs), particularly aimed at safeguarding retinal photoreceptor cells through neuroprotection. Numerous potential therapeutic candidates have been suggested, with the glial-cell-derived neurotrophic factor (GDNF) emerging as a promising option to rescue photoreceptor cells [18]. 

In our previous work, we fabricated an injectable and terminable collagen–alginate composite encapsulated cell therapy (CAC ECT) gel, encapsulating GDNF-producing human embryonic kidney (HEK) cells (Figure 1A). Although the presence of cell attachment ligands in collagen promotes cell–matrix interaction, it displays limited matrix stiffness. Therefore, alginate, being a widely utilized material for cell encapsulation [19,20,21,22] with good biocompatibility, limited cytotoxicity, and ensured quality [20,21,23], was added to form a semipermeable membrane and/or matrix. Alginate itself does not contain cell-binding motifs, which can potentially compromise cell viability over an extended period of time [24]. The combination of collagen with alginate allows the collagen–alginate composite (CAC) gel to better mimic the in vivo environment, providing mechanical protection to enclosed cells and enhancing mechanical stiffness [25,26]. Our earlier studies demonstrated that the CAC ECT gel showed promising therapeutic potential in rescuing photoreceptor cells in dystrophic Royal College of Surgeons rats, a rat model of photoreceptor degeneration [27]. In addition, a biosafety switch, utilizing Tet-on pro-Casp8, was established into the CAC ECT gel to enable gel functionality termination through oral Dox administration for a safe intraocular drug delivery (Figure 1B). However, the translation of the ECT system from the laboratory to clinical application remains elusive. Use of the rodent eye in research has been criticized due to rodents having smaller eyes compared to that of humans, anatomical difference and species difference between rodents and humans in drug pharmacokinetics, and the limited vitreous capacity of the rodent eye [28]. The rabbit is a commonly used animal model in ophthalmic studies due to its eye size being similar to that of humans, as well as their ease of handling, availability, and low cost [29,30]. Moreover, compared to rodents, rabbit eyes have enough space to accommodate multiple ocular implants, making it possible to evaluate multiple gel performance and therapeutic efficacy in a clinical situation. 

Sodium iodate (SI) is a small molecule that has been widely used for inducing retinal degeneration. It selectively damages retinal pigment epithelium (RPE) and subsequent photoreceptor cell death through oxidative stress. The SI-induced retinal degeneration model is advantageous because it reliably mimics dry AMD. Additionally, it offers a rapid and reproducible onset of damage, making it a versatile and cost-effective tool for testing potential drug treatments [31,32]. With the aim of translating the CAC ECT gel to clinical application, this study investigated (1) the in vivo safety of the CAC ECT gel, (2) the in vivo performance of the CAC ECT gel, (3) the in vivo termination of the CAC ECT gels by oral administration of doxycycline (Dox) in healthy New Zealand White (NZW) rabbits, and (4) the in vivo therapeutic potential of the CAC ECT gel in NZW rabbits with retinal degeneration induced by intravenous injection of SI. A comprehensive set of characterization methods was employed to evaluate the in vivo gel properties, as summarized in Appendix A.

## 2. Results and Discussion 

### 2.1. In Vivo Evaluation of the Effects of CAC ECT Gels on Retinal Structure and Function and Glial Cell Reactivity

Intravitreal implantation can induce toxicity, attributable primarily to the presence of a material itself. Additionally, pharmacological agents and their metabolites, especially in scenarios involving biodegradable implants for drug administration, can further exacerbate this toxicity due to their concentration and interaction within the ocular environment. Prior to applying CAC ECT gels as an intraocular drug delivery platform for clinical application, it is crucial for us to determine the safety of intravitreal surgery and the effect of the prolonged presence of the CAC gel inside the vitreous on the eye. CAC ECT gels were injected into the vitreous of NZW rabbit eyes and left for 2 weeks. Operated control rabbit eyes that received the medium only were also evaluated to determine if the surgical procedure could induce any changes in the retina. During the study, the weight and IOP of the rabbits were examined. It was noted that there was no significant difference in the body weight of the rabbit. By contrast, IOP in rabbits with gel injection revealed a short-term decrease when compared to unoperated and operated control over 2-week monitoring (Table 1). There are many causes that can lead to ocular hypotension, including intraocular inflammation, trauma, ophthalmic surgery, and the topical or systemic administration of medication [33]. Moreover, the leakage of intraocular fluid from the surgical wounds and the decrease in aqueous humor generation can lead to ocular hypotony as well [33]. In our study, two holes were made on the cornea using a 27-guage needle to release the IOP, ensuring successful implantation. The short-term decrease in IOP observed may be attributed to the leakage of aqueous humor from the wound.

The potential for ocular surgery to induce cataract formation and retinal detachment was documented when the lens and retina were disturbed. Furthermore, medication-induced cataracts are also common. BIO was employed to monitor corneal and lens opacities, the presence of cataract formation, and retinal detachment following gel implantation. Over a 2-week monitoring period, no corneal and lens opacification, cataract formation, or retinal detachment was observed, suggesting that the surgical procedure did not disturb the cornea, lens, or retina. 

The effect of the CAC ECT gel on retinal function over the two-week periods was evaluated using ERG (Figure 2A), which is a clinical method valued for its non-invasive measurement of the electrical activity of retinal neurons in response to light. The a-wave indicates the function of photoreceptor cells, and the b-wave corresponds to the collective function of ON-bipolar and Müller cells. Both a- and b-wave amplitudes were obtained from the photopic and scotopic ERG data at different light intensity settings. The a-wave and b-wave amplitude, observed in the gel-treated rabbits (three-gel and six-gel) after 2 weeks, showed no significant difference when compared with the unoperated control, highlighting that the surgical procedure and presence of the CAC ECT gel in the vitreous did not affect the retinal function.

We next asked whether intravitreal implantation of the CAC ECT gel has any toxic effect on retinal cytoarchitecture. H&E-stained retinal sections of rabbit eyes at 2 weeks after intravitreal gel injection were examined. All retinal layers were present, and retinal architecture was found to be well organized and with distinct layers, including ONL, OPL, INL, IPL, and GCL, in all operated eyes, including the operated control, three-gel group, and six-gel group when compared to the unoperated eyes (Figure 2B). A comparison of various retinal layer thickness revealed no significant differences among the operated eyes and unoperated eyes (Figure 2C). Specifically, the ONL thickness was 37.10 ± 1.64 µm in the three-gel group, 35.19 ± 1.04 µm in the six-gel group, and 39.16 ± 0.64 µm in the operated control. The OPL thickness was 9.69 ± 0.67 µm in the three-gel group, 10.20 ± 0.42 µm in the six-gel group, and 11.08 ± 0.87 µm in the operated control. Similarly, the INL thickness was 22.50 ± 1.01 µm in the three-gel group, 22.29 ± 0.76 µm in the six-gel group, and 21.62 ± 0.81 µm in the operated control, while the IPL thickness was 24.61 ± 1.64 µm in the three-gel group, 21.87 ± 0.45 µm in the six-gel group, and 21.89 ± 2.22 µm in the operated control. The results demonstrated that rabbits could endure multiple intravitreal administrations of CAC ECT gels without adverse effects, as well as without eliciting detectable toxicity within the retinal environment. Additionally, the procedure associated with intravitreal implantation, as well as the CAC gels themselves, did not cause retinal edema or neuron atrophy. Overall, the CAC gels demonstrated excellent biocompatibility and were well tolerated when applied intraocularly—results that are consistent with the findings of other studies [27,34].

In response to foreign body biomaterials, various retinal glial cells, namely microglia, astrocytes, and Müller cells, can exhibit reactivity [35]. The sustained activation of these cells is potentially harmful to the retina, leading to the dysfunction and degeneration of cells [36]. Therefore, we next investigated the effects of surgical procedures and the presence of the CAC ECT gel on these cells. Immunohistochemistry was performed on the retinal sections. Iba-1 was used to mark microglial cells, while GFAP was detected in the visualization of astrocytes and Müller cell end feet.

Microglial cells play a crucial role in preserving the stability of the retinal environment. They become active in response to changes in the parenchyma, altering their shape [37]. In all examined conditions, no significant alterations were observed in the number, distribution, or shape of the microglia (Figure 3A, Appendix A). Iba-1 immunoreactivity is primarily observed within the inner regions of the retina, such as in the ganglion cell layer (GCL), inner plexiform layer (IPL), and inner nuclear layer (INL), where inactive microglia can be seen. Once microglial cells become active, their cell bodies become larger, and the cell processes become thicker. The operated rabbit group all exhibited thin and slender projections under iba-1 immunostaining, suggesting that intravitreal implantation of the CAC ECT gel and the surgical procedure did not trigger the transformation of microglia from a dormant to an active state.

We next investigated whether intravitreal surgery and prolonged gel treatment could induce retinal stress. The expression of GFAP, a glial protein that is upregulated during gliosis, was assessed in the unoperated eyes and operated eyes (Figure 3B). In the unoperated control eyes, GFAP immunoreactivity was predominantly localized to astrocytes in the GCL. By contrast, retinas from the operated control, three-gel, and six-gel rabbits exhibited GFAP expression in astrocytes in GCL, as well as in Müller cells that traverse various retinal layers. Yet no significant difference was observed among these groups (Appendix A). Collectively, these results demonstrated that intravitreal surgery imposed a minimal increase in glial activation.

In the unoperated eye, GFAP immunoreactivity was predominantly detected in GCL. GFAP immunoreactivity was also found in Müller cells in some of the retinas of the operated control rabbits and gel-injected rabbits. Reactive gliosis of Müller glia is considered a retinal response to the injury [38,39]. Once the retina undergoes injury or stress, Müller glial cell is found to be reactive, characterized by the increase in GFAP levels [40]. Studies have indicated that the upregulation of GFAP expression could serve as a biomarker for disrupted retinal homeostasis [41]. Enhanced GFAP fluorescence in Müller cells was detected at 2 and 4 weeks after injection of poly(lactic-co-glycolic acid) (PLGA) microspheres, with a return to baseline levels following 12 and 24 months post injection [42]. Similar observations have been reported, indicating that poly(ε-caprolactone) (PCL) implants could cause retinal stress, which can be inferred from the elevated levels of GFAP [43]. The controversy revolves around whether gliosis is detrimental or advantageous to retinal tissue. Müller cell gliosis and chronic gliosis were found to accelerate disease progression and neurodegeneration. One the other hand, gliosis of Müller cells can stimulate the release of neurotrophic factors, thereby preserving retina neurons [44]. Our results demonstrated that Müller cells were reactive with minimal increase in GFAP expression following intravitreal surgery. This may be a response to disrupted retinal homeostasis caused by the invasive surgical procedure, as a similar increase in GFAP was noted in the operated control as well. 

### 2.2. In Vivo Performance of the CAC ECT Gel 

Having ascertained the safety of the intravitreal implantation of the CAC ECT gel in rabbit eyes, we next elucidate the in vivo performance of these CAC ECT gels, which is crucial for the success of the ECT system. Healthy NZW rabbits were injected three or six units of gels intravitreally. The gels were then retrieved 2 weeks after for the assessment of gel morphology, system stability, resistance to degradation, encapsulated cell viability, and GDNF release kinetics. 

#### 2.2.1. Gel Morphology, Encapsulation Power, Mechanical Stability, Material Degradation, and Internal Structure

The morphology of retrieved gels after 2-week implantation was compared with those on the fabrication day. Following fabrication, the gels displayed a distinct cell core region predominantly occupied by HEK cells, surrounded by an acellular outer region (Figure 4A). After 2-week implantation, a similar cell core area, as well as an acellular outer region, were identified in the retrieved gel (Figure 4A). It is worth noting there was no host tissue ingrowth, and there was an absence of pericapsular fibrotic overgrowth (PFO) on the gel surface. It is reported that some ECT systems had uncontrollable cell growth kinetics that led to the formation of cell protrusion, which can induce immune response, encapsulated cell necrosis, and even device failure [19]. Yet no cell protrusion was identified on retrieved gels, suggesting that CAC ECT gels display great encapsulation power. In our previous study, we found that using a 2-stage protocol to crosslink the CAC ECT gel, where collagen was crosslinked first followed by alginate gelation, led to a greater tendency for cells to undergo condensation into the gel core compared to the 1-stage protocol, where collagen and alginate are crosslinked simultaneously [27]. This difference is attributed to the continuous contraction of collagen in the 2-stage process. Meanwhile, retrieved gels were found to have intact gel integrity, since alginate-based hydrogel can undergo slow degradation or dissolution in the presence of calcium-chelating compounds and non-gelling cation [45]. The immune cells from the host attempt to degrade the foreign biomaterial as well [46]. We next investigated the ability of CAC ECT gels to resist degradation by evaluating the change in gel diameter before and after implantation. Indeed, retrieved gels in the three-gel and six-gel groups showed no significant changes in gel diameter when compared to the pre-implantation level. The average gel diameter was 494.20 ± 6.12 µm in the three-gel group and 481.70 ± 9.42 µm in the six-gel group, which were not significantly different from the pre-implantation diameter (484.42 ± 7.44 µm) (Figure 4B). This highlights that the CAC system exhibited good resistance to degradation over 2 weeks. Analysis of the thickness of the acellular outer region before and after implantation demonstrated no significant difference, with the thickness measured at 44.40 ± 2.28 µm before implantation and 43.97 ± 2.05 µm in the three-gel and 45.49 ± 2.14 µm after implantation (Figure 4C). This is consistent with previous morphological findings, where no cell protrusion was spotted after 2-week implantation. The microstructure of the retrieved CAC ECT gels was then studied using SEM. SEM examination of the gel surfaces revealed a highly compact network where collagen fibrils and alginate were intertwined, creating a complex and three-dimensional porous structure, with no cells detected on the gel surface (Figure 4D). This finding aligns with previous observations that demonstrated the presence of an acellular outer region following 2-week implantation in morphological examinations (Figure 4A). Further SEM analysis of the core region of the retrieved gel at 2 weeks showed that cells were attached to the CAC scaffold, and colonies of round cells were encapsulated within an interpenetrating network (IPN) of collagen and alginate, suggesting that composite hydrogel provides the network for cell attachment and proliferation. 

The mechanical stability of the scaffold is in direct relation to the immunoprotective role of the ECT device and the viability of encapsulated cells [47]. The ECT device has to be mechanically stable to resist the shear forces encountered during a surgical procedure and the dynamic changes in microenvironment in the targeted site following implantation. Without good mechanical stability, a fractured device is likely to expose enveloped cells to the host, thereby causing device failure. Since no gel breakage and intact gel morphology were observed in all the retrieved gels, it was clear that the CAC scaffold could tolerate the shear forces during an intravitreal surgery procedure and the changes in a vitreous microenvironment. These results collectively suggested that our CAC scaffold carries good mechanical stability, which is in line with previous studies [27,34]. A well-designed ECT device should display good resistance to undesirable degradation so that unnecessary contact between encapsulated cells and the host tissue is avoided. Therefore, biomaterials with a slow degrading rate or non-biodegradable biomaterials are required for an ECT system. Previous studies showed that CAC gel can be degraded due to collagen degradation or alginate dissociation [48]. Matrix metalloproteinases (MMPs) are a group of zinc-dependent endopeptidases that are responsible for extracellular matrix (ECM) remodeling [49], and type I collagen can be degraded in the presence of MMP2 [50]. Various MMPs such as MMP1, MMP2, and MMP9 are found in human vitreous [51]. Meanwhile, alginate-based hydrogel undergoes dissolution or degradation in the presence of calcium-chelating compounds such as phosphate, citrate, and lactate and cations (Na^+^, Mg^2+^), and the process occurs in a slow and uncontrollable way [45]. Inorganic ions such as Na^+^, K^+^, Cl^−^, Ca^2+^, and Mg^2+^ are present in human vitreous [52]. The dissolution of CAC gel can be reflected by the gradual decrease in gel diameter. Yet our results showed that over the 2-week monitoring period, there was no visible change in gel diameter, which indicated that the CAC scaffold had good resistance to degradation. These results were consistent with those observed in earlier studies demonstrating that the CAC gel remained well resistant to degradation [27,34]. There are several possible explanations for the long-term resistance ability to degradation in ocular tissue. Firstly, the concentration of Na^+^ and Ca^2+^ is relatively low in ocular tissue [53]. Moreover, an increased MMP2 level is found to mediate vitreous liquefication, which is generally expressed in elderly people or in some pathological states such as diabetic retinopathy [54]. In fact, during vitreous sample collection, we found that the majority of collected vitreous is gel-like, which may be an indication of low MMP2 levels. However, this warrants further investigation. There are various methods available for collagen crosslinking, and each method can significantly affect the mechanical stability of the hydrogel. Meanwhile, the crosslinking method is crucial for alginate as well. Considering the requirement for a cell-friendly environment with relatively low toxicity in cell encapsulation, we chose to crosslink collagen under mild temperature conditions, followed by alginate gelation with 0.1 M CaCl_2_. According to our previous research results, this crosslinking method not only ensures the mechanical stability of the material but also provides optimal support for CAC ECT gel performance [27].

One of the major restrictions to the clinical use of alginate in an ECT system is the occurrence of a foreign body reaction (FBR), which causes the dysfunction of encapsulated cells [55]. After the implantation of a biomaterial onto the host tissue, ECM can be deposited and adhered to the implant. This process is followed by the deposition of a variety of immune cells, such as neutrophils, monocytes, macrophages, and lymphocytes, on the surface of the implant, eventually leading to the formation of fibrotic tissue. The presence of PFO around the devices or implants can restrict the diffusion of oxygen and nutrients, as well as the therapeutic drug, which can lead to a drop in the viability of encapsulated cells and device dysfunctionality [56]. A morphological examination of gels retrieved at 2 weeks showed no presence of PFO. These data indicated that CAC gels do not elicit FBR after an extended period of implantation, which is in agreement with Wang’s findings showing that alginate microspheres loaded with retinoic acid also revealed the absence of PFO following intravitreal implantation [57]. A possible explanation for this might be that the eye is an immune-privileged organ [58]. 

Moreover, a good biomaterial for the ECT system should enable encapsulated cells to proliferate inside the matrix so that cells can retain good viability and functionality. When cells are entrapped in collagen-based hydrogel, cells can adhere to collagen via integrin. Consequently, the interaction between cells and the matrix can be enhanced, thereby improving cell viability. Our SEM results at 2 weeks demonstrated that HEK cells attached to the CAC scaffold in which thick and concentrated collagen bundles were identified, and colonies of cells were entrapped in the IPN network. These findings suggested that the CAC scaffold provided a compact structure for the cells to proliferate, which is in line with the findings of previous studies [48] demonstrating that the CAC gel exhibited a rough surface and a complicated network that facilitated cell adhesion and proliferation. 

#### 2.2.2. Encapsulated Cell Viability and GDNF Secretion

The viability of the encapsulated cell is considered a critical factor affecting the application of the ECT system. After 2 weeks of implantation, CAC ECT gels were retrieved from three-gel and six-gel rabbits to study their viability. Images of the Live/Dead assay of the retrieved gels from the three-gel and six-gel groups showed the distribution of viable cells and dying cells (Figure 5A). These results suggested that encapsulated cells were mostly viable after 2 weeks of implantation. Further analysis of cell viability revealed no significant difference between the three-gel and six-gel groups in terms of cell viability (Figure 5B). These results suggest that multiple gel injections do not cause a significant reduction in cell viability and the potential for the utilization of multiple gel administrations in personalized therapeutic strategies. The vitreous GDNF level of the unoperated control, operated control, three-gel group, and six-gel group was assessed by GDNF ELISA at 2 weeks of implantation (Figure 5C). The GDNF level in the unoperated control and operated control was almost undetectable, and no significant difference was detected. For rabbits receiving three-gel and six-gel implants, accumulated GDNF levels of 2342.76 pg/mL and 4321.92 pg/mL were detected, respectively, after 2 weeks of implantation, which were significantly higher than that of the operated control. By doubling the amount of the CAC ECT gel injected into the vitreous body, we observed a corresponding two-fold positive increase in GDNF levels. These results suggested that CAC ECT gels with permeability were able to secrete GDNF into the vitreous in a dose-dependent manner. These results collectively suggested that the CAC gel was selectively permeable to allow for GDNF diffusion from the gel to the rabbit vitreous while the host immune cell could not enter the gel matrix. 

Permeability is another factor to be considered when designing ECT technology. ECT device should be designed with selective permeability to allow gas and nutrients in and out of the device and to enable the release of drugs from the device to the targeted site. SEM examination on the gel surface suggested the formation of a porous structure. Moreover, accumulated GDNF levels were detected in the rabbit vitreous. These results collectively suggested that the CAC gel was selectively permeable to allow for GDNF diffusion from the gel to the rabbit vitreous while the host immune cell could not enter the gel matrix. The viability of encapsulated cells and their proliferation inside the gel matrix is critical for the application of the ECT device. By comparing the cell viability between the three-gel and six-gel groups, we found that the number of hydrogels implanted into the vitreous does not significantly affect the viability of the encapsulated cells and that the levels of GDNF are closely associated with the quantity of implants, which suggested that encapsulated cells were functional and secreted GDNF into the vitreous, owing to gel permeability. The selective permeability of the CAC ECT system can be explained through several aspects. Firstly, a SEM examination of the gel surface revealed the presence of a porous structure, which serves as the physical basis for selective permeability. This structure allows for the free diffusion of oxygen, nutrients, and therapeutic molecules such as GDNF, while preventing larger immune cells from entering the gel matrix. Additionally, the detection of accumulated GDNF levels in the rabbit vitreous indicates that the CAC gel effectively controls the release of GDNF while preventing the intrusion of harmful external factors, thereby ensuring therapeutic efficacy. Studies have shown that the number of gels implanted into the vitreous does not significantly affect the viability of the encapsulated cells, suggesting that the gel’s permeability is sufficient to maintain cell survival and function. Moreover, the level of GDNF secretion is closely associated with the number of implants, further confirming the effectiveness of the gel’s permeability. Lastly, the selective permeability of the CAC gel plays a crucial role in immune protection by restricting the entry of immune cells, thereby preventing the host immune system from attacking the encapsulated cells and decreasing ECT device failure. As discussed before, an individual may require different dosages to meet therapeutic efficacy, and this can be achieved by tuning the number of implants. Together with previous findings, we envision the feasibility of employing the CAC ECT gel for personalized medicine, which can be tuned by the number of gel implants.

### 2.3. In Vivo Termination of the CAC ECT Gel 

One major challenge is that when undesirable adverse effects are noted by the physician or the final goal is achieved, the necessary removal of implants is problematic [59]. The development of a biosafety switch that enables the termination of device functionality without the requirement of invasive surgical removal is essential. Nevertheless, limited ECT systems have been designed with this “biosafety switch”. In order to tackle this problem, previous work by our team proposed a Tet-on pro-Casp8 system as a promising alternative for ECT system termination. GDNF-secreting HEK cells, equipped with the Tet-on pro-Casp8 system, are capable of undergoing apoptosis upon administration of doxycycline, regardless of whether the cells are in a non-proliferative phase [27]. 

The morphology of gels retrieved from Dox-treated rabbits was compared with those retrieved from rabbits without Dox treatment. All retrieved gels were found to have intact gel integrity, no gel breakage, and good encapsulation power in morphological studies (Figure 6A). No cell leakage was observed in both groups (Figure 6A). When the gel surface was examined, there was an absence of PFO, and no host tissue ingrowth or adherence were detected (Figure 6A). However, cell shrinkage and massive fragmented cell debris that were homogenously distributed in the cell core were spotted in Dox-treated gel, while enclosed cells in the non-Dox group had normal features such as a spheroid shape, smooth appearance, and larger size (Figure 6A). A Live/Dead assay demonstrated that the majority of encapsulated cells in the Dox-treated group were dead, as indicated by the red fluorescence. Only a small number of cells were identified as “living” with green fluorescence, but these cells appeared to be smaller and had a broken morphology compared to those in the non-Dox-treated gels (Figure 6B). A decrease in cell viability of up to 95% was detected in the Dox-treated gels determined by an MTS assay (Figure 6C), which was consistent with the Live/Dead analysis (Appendix A). The secreted GDNF level in the vitreous of the non-Dox and Dox groups was assessed by GDNF ELISA at 1 week after Dox administration. Compared with the non-Dox treatment group, the rabbits receiving Dox treatment showed a significant decrease in accumulated GDNF levels (Figure 6D), which further confirms that Dox could effectively terminate gel functionality. 

Recent studies first showed retinal toxicities of Dox via the intravitreal route using NZW rabbits [60]. To determine whether Dox administration imposed toxicity on retinal function, in vivo retinal function was assessed using ERG after a 1-week Dox administration in healthy NZW rabbit eyes. Both Dox-treated and non-Dox-treated groups experienced minor alterations in photopic and scotopic a-wave and b-wave responses, with no statistically significant differences observed (Figure 6G). These results suggested that the function of both cone and rod photoreceptors, as well as inner retinal cells, was not significantly affected by Dox administration. To assess whether the 1-week oral administration of Dox had any effect on retinal architecture, retinal cytoarchitecture was assessed by H&E histology. Under gross examination, the retinal cytoarchitecture of unoperated control, Dox-treated, and non-Dox-treated groups were all well preserved (Figure 6E). Analysis of the thickness of various retinal layers revealed no significant differences among the unoperated control, non-Dox-treated group, and Dox-treated group (Figure 6F). Specifically, the thickness of the ONL was 36.24 ± 1.60 µm in the unoperated control, 37.10 ± 1.64 µm in the non-Dox-treated group, and 37.43 ± 0.59 µm in the Dox-treated group. The OPL thickness was 11.26 ± 0.96 µm in the unoperated control, 9.69 ± 0.67 µm in the non-Dox-treated group, and 9.33 ± 0.43 µm in the Dox-treated group. Similarly, the INL thickness was 18.40 ± 1.55 µm in the unoperated control, 22.50 ± 1.01 µm in the non-Dox-treated group, and 20.29 ± 2.30 µm in the Dox-treated group. The IPL thickness was 25.63 ± 0.45 µm in the unoperated control, 24.61 ± 1.64 µm in the non-Dox-treated group, and 23.67 ± 1.44 µm in the Dox-treated group. The GCL thickness was 20.05 ± 1.73 µm in the unoperated control, 16.80 ± 1.22 µm in the non-Dox-treated group, and 19.40 ± 1.49 µm in the Dox-treated group. These results suggest that oral Dox administration imposed no detrimental damage to retinal architecture.

Dox can be administrated to the eye through many routes, such as oral [61], topical [62], intravitreal [60], intravenous [63], intraperitoneal [64], subretinal [65], and subcutaneous [66] administration. We have chosen to use the oral route in our studies, as it can mimic clinical situations with human subjects. In this study, 0.1 mg/mL of Dox in drinking water was applied to trigger the termination of the CAC gel. Our results showed that Dox crossed the blood–retinal barrier and significantly decreased the viability of encapsulated cells. Only a small number of cells were identified as “living” with green fluorescence, but these cells appeared to be smaller and had a broken morphology compared to those in the non-Dox-treated gels, suggesting that these cells may not be viable, which was consistent with the MTS assay. These data were also consistent with the GDNF analysis on the vitreous level, demonstrating that gel functionality was almost terminated, with undetectable GDNF levels. Ejstrup et al. assessed the half-life of GDNF, which is around 37 h in the porcine eye [67]. Due to the smaller volume of rabbit vitreous, it was expected that drug clearance would be more rapid in rabbits. These results were in accordance with our previous in vitro studies indicating that 1-day treatment of Dox could effectively induce cell apoptosis, and 3-day treatment can inactivate gel functionality in rat eyes [27]. The dosage of Dox required to inactivate gel functionality in rabbits was comparable to the results from earlier studies on rats, where 1 mg/mL of Dox in drinking water could terminate gel functionality; this indicated that Tet-on pro-Casp8 was highly sensitive to Dox. Further work is required to identify the lowest concentration via oral administration that can trigger the inactivation of gel functionality. 

Retinal function and architecture were not affected by oral administration of Dox, as evidenced by ERG examination and H&E histology. Meanwhile, no fundus abnormalities or cataract formation were detected in the rabbits with terminated gels. These data were in agreement with previous findings, which demonstrated that a lower dosage of Dox via intravitreal administration has no toxicity to retina [61]. Recent studies have reported that, similar to many tetracyclines, Dox is known to have an anti-inflammatory effect, which is achieved through the inhibition of tumor necrosis factor-alpha (TNF-α) and MMPs. Dox has exhibited neuroprotective effects in many animal models with neurodegeneration [68]. Dox was also found to have beneficial effects for diabetic retinopathy through its anti-inflammatory effect [69]. These findings, together with our results, confirmed the safety of oral administration of a low dosage of Dox on eye tissue to terminate gel functionality. 

### 2.4. Therapeutic Efficacy of the CAC ECT Gel in Rabbits with Retinal Degeneration 

To demonstrate the ability of CAC ECT gels to secrete GDNF to treat a retinal degenerative disease state, a SI-induced model of retinal degeneration in a rabbit eye was employed. SI first directly damages RPE cells, causing secondary photoreceptor cell death, which serves as a good model for studying the novel treatments for photoreceptor cell rescue [70]. An intravenous injection of SI was performed on rabbits to establish a retinal degeneration model, followed 2 weeks later by an intravitreal injection of three or six units of the CAC ECT gel into the rabbit vitreous to assess its therapeutic potential. Rabbits without SI injection served as the baseline; a- and b-wave amplitudes of rabbits with retinal degeneration receiving no gel, three-gel, and six-gel treatment were measured by ERG 2 weeks after intravitreal gel injection. The amplitude of the a-wave in SI-treated rabbits without gel injection was significantly lower than the baseline level, with a measured value of 75.35 ± 6.04 µV compared to the baseline level of 92.62 ± 9.16 µV (Figure 7A). This reduction was alleviated by three-gel and six-gel treatment, with a-wave amplitudes of 85.88 ± 6.08 µV and 89.99 ± 8.88 µV, respectively, for the three-gel and six-gel groups. Similarly, both the three-gel and six-gel groups exhibited significant recovery in scotopic b-wave amplitude compared to the SI-only group, which had a b-wave amplitude of 144.78 ± 8.85 µV. The three-gel group showed a b-wave amplitude of 157.78 ± 8.58 µV, while the six-gel group demonstrated an even greater recovery with a b-wave amplitude of 165.79 ± 8.88 µV. After ascertaining the beneficial effects of the CAC gel on retinal function, we proceeded to evaluate whether the CAC ECT gel could rescue photoreceptor cells by an H&E staining assessment (Figure 7B). Compared to the baseline level, the rabbits that received the SI injection without gel implantation exhibited a marked reduction in ONL thickness, with an average thickness of 30.37 ± 3.75 µm compared to the baseline thickness of 37.46 ± 3.78 µm (Figure 7C). Similarly, the number of ONL nuclei per mm of retina was significantly reduced to 1337 ± 134 nuclei/mm, compared to the baseline count of 1856 ± 65 nuclei/mm (Figure 7D). However, both the three-gel and six-gel treatment groups displayed remarkable increases in ONL thickness, with the three-gel group showing an average thickness of 38.49 ± 4.20 µm and the six-gel group showing 38.34 ± 5.49 µm (Figure 7C). The number of ONL nuclei per mm also increased significantly when compared to the no-gel-treated rabbits, with the three-gel group exhibiting 1843 ± 60 nuclei/mm and the six-gel group reaching 1893 ± 107 nuclei/mm (Figure 7D). These findings suggest that the CAC ECT gel was effective in rescuing photoreceptors in the rabbit model of SI-induced retinal degeneration.

To assess whether the CAC ECT gel had potential in protecting photoreceptor cells from apoptosis, retinal sections from healthy rabbits and SI-injected rabbits with or without gel treatment were stained with TUNEL (Figure 7E). A notable increase in the number of TUNEL-positive apoptotic cells in the ONL was observed in the SI-treated group that did not receive gel treatment (121 ± 35 cells per mm) compared to the baseline (10 ± 2 cells per mm). However, the number of TUNEL-positive cells in the ONL significantly decreased in both the three-gel (26 ± 4 cells per mm) and six-gel (20 ± 4 per mm) treatment groups compared to non-gel-treated rabbit retinas (Figure 7F). These data suggested that three units of the CAC ECT gel treatment were highly effective in mitigating photoreceptor apoptosis in the retina. To explore whether functional and morphological rescue was attributed to GDNF secreted from the CAC ECT gel, the gels were retrieved from the rabbit vitreous for the examination of their overall structure and cell viability. A Live/Dead assay demonstrated that the encapsulated cells were mostly viable after 2-week implantation, with an intact gel boundary and no cell leakage (Figure 7G). Further analysis of cell viability revealed no significant differences between the three-gel and six-gel groups (Appendix A). As for the GDNF level, a similar GDNF release pattern was observed in rabbits with retinal degeneration compared to healthy rabbit eyes (Figure 7H). These findings indicated that CAC ECT gels released GDNF into the vitreous in a dose-dependent manner, leading to photoreceptor rescue in rabbits with retinal degeneration.

The neuroprotection of photoreceptor cells against degeneration is a promising alternative strategy to treat RRDs characterized by photoreceptor loss and degeneration. Our results showed that 3- and six-gel treatment confer neuroprotection to photoreceptor cells against retinal degeneration induced by SI. Beneficial effects in retinal function were found in rabbits with three-gel and six-gel treatment compared to no-gel-treated rabbits. We then evaluated the ONL thickness and ONL nuclei per mm, since a thinner ONL layer reflects photoreceptor loss [71,72]. Our results demonstrated the thinning of ONL and a decrease in the ONL nuclei number after SI injection, as previously reported [31]. Rabbits treated with three-gel and six-gel injections showed thicker ONL and an increase in the number of nuclei in ONL than no-gel-treated rabbits, which suggested that the CAC ECT gel can rescue photoreceptor cells. To further confirm whether functional and morphological rescue was attributed to the CAC ECT gel, we retrieved our gel and collected the vitreous from SI-induced rabbits to assess encapsulated cell viability and vitreous GDNF levels. These data demonstrated that encapsulated cells were mostly viable and released GDNF in a dose-dependent fashion. These data collectively demonstrated that the CAC ECT gel can secrete bioactive GDNF to improve retinal function and rescue photoreceptor cells, which corroborates the findings of our and other studies indicating that employing the ECT system to deliver GDNF is a promising strategy in the management of chronic degenerative or neurologic disorders [27,73,74]. Compared to the normal eye condition, the environment of degenerative retinas may be less favorable for ECT system performance. Yet CAC ECT gels remained viable in degenerative eyes, suggesting that CAC ECT gels have the potential to treat retinal degeneration. SI was found to induce photoreceptor death by apoptosis in animals [75], which is a hallmark pathological feature in most RDDs, like AMD and RP [76]. Our observations revealed a significant increase in the number of TUNEL-positive cells in the ONL following SI treatment, consistent with previous findings in mice [77] and rabbits [78], indicating that SI-induced retinal degeneration could serve as a reliable model for studying RDDs. Most importantly, gel-treated rabbits showed a significant decrease in the number of TUNEL-positive nuclei in ONL. Together with the findings on cell viability in retrieved gels and increased accumulated vitreous GDNF levels, it is apparent that GDNF released from CAC ECT gels was able to reduce the number of TUNEL-positive cells in ONL. These results indicated that GDNF released from the CAC ECT gel was able to prevent apoptosis in photoreceptor cells. Our data also demonstrated that three units of CAC gels were able to have promising therapeutic potential in improving retinal function and morphological rescue. Although there are currently formulas available to convert animal doses to human-equivalent doses and vice versa [79], their applicability in our study requires further investigation when considering the fact that our study focuses on intravitreal administration. As discussed previously, individual patients may require different dosages to achieve a therapeutic outcome; this may be achieved by tuning the number of implants. 

However, the number of CAC ECT implants required to meet the needs of patients in different disease stages has not been properly studied and will require further investigation. 

## 3. Materials and Methods

### 3.1. Cell Culture of HEK/293/GDNF/Tet-on pro-Casp8

HEK 293 cells, engineered to overexpress GDNF and equipped with a Tet-on pro-Casp8 biosafety switch, were cultured as previously described [27]. The cell line was maintained in Dulbecco’s Modified Eagle Medium with high glucose (DMEM-HG, Life Technologies, Carlsbad, CA, USA) supplemented with 10% Tet-on™-approved fetal bovine serum (Clontech, Mountain View, CA, USA) and 1% penicillin/streptomycin (Life Technologies, Carlsbad, CA, USA). To sustain the overexpression of GDNF and the functionality of the Tet-on pro-Casp8 system, the culture medium was further supplemented with 7.5 μg/mL blasticidin (Invitrogen, Carlsbad, CA, USA), 300 μg/mL zeocin (Invitrogen, Carlsbad, CA, USA), and 500 μg/mL G418 sulfate (Gibco, Grand Island, NY, USA). The supplemented medium was refreshed every three days. 

### 3.2. Preparation of the CAC ECT Gel 

The CAC ECT gel was fabricated as previously described [28]. HEK/293/GDNF/Tet-on pro-Casp8 cells were trypsinized with 0.25% Trypsin-EDTA (Thermo Fisher Scientific, Waltham, MA, USA). The HEK cell pellet was collected and then mixed with a rat-tail neutralized collagen type I (BD Bioscience, Franklin Lakes, NJ, USA) solution in the presence of 1N NaOH and a sterilized 1.5% w/v alginate solution (Sigma-Aldrich, Burlington, MA, USA) to form a cell–CAC mixture. A 2-stage protocol was applied to fabricate the CAC gel in this study. In brief, 2 μL aliquots of the cell–CAC mixture were injected into glass molds (Sigma-Aldrich, Cat. P0674, Burlington, MA, USA) and then incubated at 37 °C to initiate collagen gelation for 75 min, followed by immersion in 100 mM CaCl_2_ in a 0.9% NaCl bath for alginate gelation. After 75 min, the CAC ECT gels were collected from the glass molds and rinsed with 0.9% NaCl (Figure 8). The CAC ECT gels were cultured using 24-well plates before conducting intravitreal gel injection in rabbits. In general, each gel was in a cylindrical shape, approximately 7 mm in length and 480 μm in diameter, and contained 5 × 10^4^ GDNF-secreting HEK cells.

### 3.3. Animal Care and Use 

Female NZW rabbits, aged three to four months and weighing 2.8 to 3.1 kg, were housed in a room with controlled temperature and subjected to a 12 h light/dark cycle. All animal handling procedures and experiments were conducted in line with the Association for Research in Vision and Ophthalmology (ARVO) Statement for the Use of Animals in Ophthalmic and Vision Research. The use of live animals for research and teaching purposes was approved by the Faculty Committee at The University of Hong Kong (CULATR No. 4927-19 and CULATR No. 5835-21). All researchers involved in the assessment of outcomes, including those performing the procedures, monitoring, and analyzing the results, were blinded to the treatment groups to minimize bias. The code for group assignment was only broken after the completion of data analysis.

### 3.4. Intravitreal Gel Injection 

NZW rabbits were anesthetized by an intramuscular injection of a mixture of xylazine (5 mg/kg) and ketamine (45 mg/kg) (Alfasan lab, Woerden, Zuid-Holland, The Netherlands) as previously described [80]. Gels were intravitreally injected into their eyes as previously reported [27]. In brief, the right eyelid and surrounding ocular tissue were first cleaned with a diluted betadine solution (Mundipharma A.G., Frankfurt, Germany) to ensure sterility. Topical analgesia was achieved by applying 0.5% Alcaine (Alcon, Belmont, CA, USA) onto the cornea. Afterwards, 1% Mydriacyl (Alcon) was applied onto the cornea to dilate the pupil. The conjunctiva was then separated from the sclera with the help of Vaness curved scissors. One incision (less than 2 mm from the limbus) was then carefully made into the vitreous with a 20-gauge blade (Alcon, Belmont, CA, USA) under a surgical operating microscope (Zeiss) (Figure 9). Retinal and lens damage was avoided. To release enhanced intraocular pressure (IOP) during intravitreal gel injection, two holes were made near the corneal limbus using a 27-gauge needle. Three or six units of CAC ECT gels, suspended in a DMEM-HG 4-(2-hydroxyethyl)-1-piperazineethanesulfonic acid (HEPES) medium (Gibco, Grand Island, NY, USA), were injected into the vitreous through sterilized round 0.5 mm thick gel-loading pipette tips (Corning, GA, USA). Upon completion of intravitreal gel injection, the wound was closed using 8-0 nylon suture. Finally, a subconjunctival injection of 2 mg/kg gentamicin (Vetone, Sydney, New South Wales, Australia) was given. A topical administration of antibiotic ointment (Tobrex, Fort Worth, TX, USA) was applied onto rabbit cornea for seven consecutive days to prevent post-operative infection. For rabbits serving as the operated control, one incision into the vitreous was made with an intravitreal injection of the same volume of the DMEM-HG HEPES medium. 

### 3.5. IOP, Binocular Ophthalmoscope (BIO) Examination, and Body Weight Measurement

Prior to any surgical intervention, the animals underwent a 2-week training period to acclimatize them to the IOP measurement process. The assessment of IOP was conducted utilizing Tonovet Plus (iCare, Vantaa, Finland), with an average of six readings taken for each measurement. To minimize the effect of circadian rhythm on IOP, the measurement of IOP was assessed at 5 p.m. [81]. IOP measurements were performed before surgery and at 2 weeks post-injection. After IOP measurement, BIO (All Pupil Indirect; Keeler, Malvern, PA, USA) was used to monitor corneal and lens opacities, as well as the presence of cataract formation and retinal detachment. For body weight measurement, the rabbit was kept calm and was gently placed on the weighing pan. Its weight was recorded to two decimal places using a digital electronic balance. 

### 3.6. ERG Recording

The rabbits were anesthetized by an intramuscular injection of a mixture of xylazine (5 mg/kg) and ketamine (45 mg/kg) (Alfasan lab, Woerden, Zuid-Holland, The Netherlands) [80]. For the assessment of retinal function after operation, ERG measurements were conducted at around 2:00 p.m. to mitigate the impact of possible circadian variations. The rabbits were given 0.5% proparacaine hydrochloride ophthalmic drops (Alcon, Fort Worth, TX, USA) for topical anesthesia. Then, 1% tropicamide (Alcon) was applied onto rabbit cornea to dilate the pupil. Artificial tears (Tears Naturale II, Fort Worth, TX, USA) were applied onto the cornea to moisturize the cornea for better signal conduction. Before photopic ERG studies, the rabbits underwent 20 min of light adaptation. The photopic ERG responses were assessed using Espion E2 Electrophysiology System at 5.0 cd.s/m^2^. The final photopic ERG responses were averaged from fifteen consecutive measurements. The rabbits then underwent dark adaptation in total darkness for at least 10 min. ERG responses to a flashlight stimulus at 0.01 and 10.0 cd.s/m^2^ were recorded. The amplitudes of the a-wave and b-wave were collected for each photopic and scotopic stimulus. 

### 3.7. Retinal Morphological Examination and Analysis

The rabbits were euthanized via intravenous injection of sodium pentobarbital (100 mg/kg; Alfasan, Woerden, Zuid-Holland, The Netherlands) as previously described [80]. Their eyeballs were enucleated and fixed with 4% paraformaldehyde (PFA) (Sigma-Aldrich, Burlington, MA, USA) in phosphate-buffered saline (PBS) overnight. The superior and inferior portions of the fixed eyeballs were then removed, leaving a 10 mm wide central section with the optic nerve attached for paraffin wax embedding (TissuePrep™ Embedding Media, Thermo Fisher Scientific, Waltham, MA, USA at 60 °C. The paraffin-embedded eye samples were horizontally cut into 5 μm thick sections using a microtome (Microm HM 315R). The sections containing the optical nerve were stained with hematoxylin and eosin (H&E) staining and photographed with an upright microscope (Eclipse 80i, Nikon, Shinagawa, Tokyo). The thickness of various retinal layers was measured using ImageJ software (1.53k). 

### 3.8. Immunohistostaining

After de-paraffinization and rehydration, retinal sections were subjected to a 10-min room temperature incubation with proteinase K in PBS (1:500) for antigen retrieval and followed by a 1-h incubation with 10% goat serum (Vector Laboratories, Mowry Ave, CA, USA) in a primary diluent to prevent unspecific binding. The sections were then incubated with an anti-glial fibrillary acidic protein (GFAP) antibody (1:500; Cat# 14-9892-82, ThermoFisher, Waltham, MA, USA) and anti-ionized calcium-binding adaptor molecule 1 (Iba-1) antibody (1:500, catalog # 019-19741, FUJIFILM Wako Pure Chemical Corporation, Osaka, Japan) overnight at 4 °C, and later with Alexa Fluor 568 goat anti-mouse IgG (Invitrogen, Cat# A-11004, Waltham, MA, USA) and Alexa Fluor 488 goat anti-rabbit IgG (Invitrogen, Cat# A-11008, Waltham, MA, USA) for 1 h at room temperature. Negative controls involved substituting the primary antibody with a blocking buffer (10% goat serum). All images were captured with a fluorescence microscope (Eclipse 80i; Nikon) equipped with a digital camera (Diagnostic Instruments, Inc., Sterling Heights, MI, USA) using a 20× objective. GFAP fluorescence intensity was quantified using ImageJ software as previously described [82]. Briefly, rectangles were drawn from the apical surface of the ganglion cell layer to the retinal pigment epithelium, and the average GFAP fluorescent intensity in these rectangular areas was measured.

### 3.9. In Vivo Gel Safety Study 

In this study, NZW rabbits were randomly assigned into 3 groups, each consisting of at least 3 animals for the in vivo gel safety study. The groups were designated as follows: Group 1 served as operated control and received the intravitreal gel procedure without gel treatment; Group 2 received an intravitreal injection of three units of gel; and Group 3 received an intravitreal injection of six units of gel. Following the administration of the intravitreal gel procedure and either three or six units of gel into the eyes, the rabbits were monitored over a 2-week period. Retinal detachment and cataract formation were assessed using a binocular ophthalmoscope to evaluate the structural integrity of the eye. Retinal function was measured via ERG to detect any functional impairments. After the 2-week observation period, the rabbits were sacrificed, and their eyes were collected for further analysis. Retinal stress markers were evaluated using immunohistochemistry (IHC), while retinal morphology was assessed through hematoxylin and eosin (HE) staining to examine any histological changes (Figure 10A).

### 3.10. In Vivo Gel Performance Study 

In this study, healthy NZW rabbits were randomly divided into four groups of at least three animals each for the in vivo gel performance study. Group 1 served as the unoperated control without any treatment; Group 2 served as the operated control and received an intravitreal gel procedure without gel treatment; Group 3 served as the three-gel group, receiving an intravitreal injection of three units of gel; and Group 4 received an intravitreal injection of six units of gel. The rabbits were sacrificed 2 weeks post injection. Gels were collected from the rabbit vitreous for their morphological examination, gel thickness analysis, assessment of cell viability and proliferation, and examination of internal microstructure (Figure 10B). In addition, vitreous fluid was extracted and preserved at −80 °C for subsequent measurements of the accumulated GDNF levels (see below).

### 3.11. In Vivo Gel Termination Study 

In this study, healthy NZW rabbits were randomly divided into two groups of at least four animals each for the in vivo gel termination study. Healthy NZW rabbits receiving three units of gels were given drinking water containing 0.1 mg/mL Dox (Vetafram) supplemented with 3% sucrose (protected from light) for up to 1 week, starting 1 week post injection. The rabbits were sacrificed 2 weeks after intravitreal gel injection, and CAC ECT gels were retrieved from the rabbit vitreous for the examination of cell viability and morphology (Figure 10C) (see below).

### 3.12. In Vivo Therapeutic Potential of the CAC ECT Gel 

In this study, rabbits receiving an intravenous injection of sodium iodate (SI) were randomly divided into three groups of at least six animals each for the in vivo therapeutic potential study. Group 1 served as the no-gel control receiving an intravenous SI injection without gel treatment. Group 2 served as the three-gel group receiving an intravenous SI injection and intravitreal injection of three units of gel, and Group 3 received an intravenous SI injection and intravitreal injection of six units of gel. Healthy NZW rabbits without any treatment served as the baseline. An SI-induced retinal degeneration model was established as previously described [83]. SI powder (Sigma-Aldrich, Cat# S4007-100G, purity ≥ 99%, Burlington, MA, USA) was dissolved in 0.9% NaCl, which was then sterilized using a syringe filter (Millex-GP, Sigma-Aldrich, Burlington, MA, USA). Upon induction of general anesthesia by an intramuscular injection of a mixture of xylazine (5 mg/kg) and ketamine (45 mg/kg) (Alfasan lab, Woerden, Zuid-Holland, The Netherlands), the SI solution was injected intravenously into the marginal ear vein of the rabbit ear. At 2 weeks post-SI-injection, three or six units of the CAC ECT gel that were fabricated one day earlier were intravitreally injected into their right eyes for the assessment of the therapeutic potential of gels (Figure 10D). 

**Figure 10 marinedrugs-22-00394-f010:**
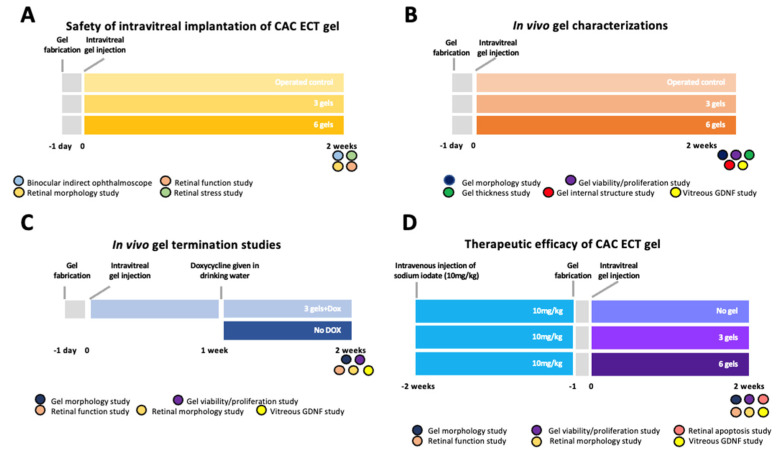
Experimental design illustrating the (**A**) in vivo safety assessment, (**B**) in vivo performance evaluation, and (**C**) in vivo termination for the CAC ECT gel in healthy NZW rabbits and (**D**) therapeutic efficacy in a sodium-iodate-induced retinal degeneration model in rabbits. CAC ECT, collagen–alginate composite encapsulated cell therapy.

### 3.13. Assessment of Cell Viability and Proliferation 

The viability of encapsulated cells in the retrieved gels was evaluated by a tetrazolium-based Cell Titer 96 Aqueous Non-radioactive Cell Proliferation assay (3-(4,5-dimethylthiazol-2-yl)-5-(3-carboxymethoxyphenyl)-2-(4-sulfophenyl)-2H-tetrazolium, inner salt (MTS) assay, Promega, Madison, WI, USA) according to the manufacturer’s instructions. The gels were collected from the rabbit vitreous, placed into the wells of 48-well culture plates, and rinsed with 0.9% NaCl. Then, a mixture of 40 μL of MTS, 4 μL of phenazine methosulfate (PMS), and 200 μL of a serum-free medium (SFM) was added. After an incubation at 37 °C for 2 h, absorbance at 490 nm was measured with a microplate reader (Elx800, Biotek, Winooski, VT, USA). For background control, a mixture of 40 μL of MTS, 4 μL of PMS, and 200 μL of SFM was added into the well without the gel. The absorbance of the retrieved gel from each experimental group was normalized with that on the day of gel fabrication. For the Live/Dead assay, the retrieved gels in the 48-well plate were rinsed with 0.9% NaCl and supplemented with 2 μM of Calcein AM and 4 μM of Ethidium homodimer-1 (Molecular probes, Waltham, MA, USA) in SFM. Subsequently, the plates were incubated for 40 min at room temperature to allow for the homogenous staining of viable and dying cells. Images of gels stained with Live/Dead assay were acquired under a confocal microscope (LSM980, Zeiss, Oberkochen, Germany) or a fluorescence microscope (Eclipse 80i; Nikon). The assessment of cell viability within the gels was conducted in accordance with the method previously described [28]. In brief, the quantification of viable and non-viable cells was performed by enumerating the pixels corresponding to live (green) and dead (red) cells in confocal microscopy images, utilizing the ImageJ software for analysis. Subsequently, cell viability was computed using the following formula:Cell viability (%) = Living cells/(Living cells + Dead cells) × 100%

### 3.14. GDNF Quantification 

The rabbit vitreous was collected, immediately frozen in liquid nitrogen, and stored at −80 °C prior to ELISA. Each vitreous sample in the form of a gel-like structure was processed to a completely liquid-like structure by sonication (Sonics&Materials, Thermo Fisher, Waltham, MA, USA). GDNF levels in the rabbit vitreous were determined using the human GDNF ELISA Kit (#EHGDNF, ThermoFisher Scientific, Waltham, MA, USA) according to the manufacturer’s instructions. The absorbance was read at 450 nm within 30 min using a multiplate reader (Elx800, Biotek).

### 3.15. Scanning Electron Microscope (SEM)

The retrieved gels were carefully washed with a 0.9% saline solution and then fixed with 4% PFA (Sigma-Aldrich, Burlington, MA, USA) at 4 °C overnight. Subsequently, the gels underwent dehydration using a graded series of ethanol solutions. After further processing at the Electron Microscope Unit (Queen Mary Hospital), critical point drying (CPD 030, Bal-tec) was carried out, and the samples were then sputtered with gold (SCD 005, Bal-tec). Finally, the samples were loaded on aluminum specimen holders for SEM imaging (LEO 1530 FEG SEM, Zeiss).

### 3.16. Terminal Deoxynucleotidyl Transferase (TdT)-Mediated dUTP Nick End Labeling (TUNEL) Assay

Retinal sections were subjected to TUNEL assay (DeadEndTM Fluorometric TUNEL System, Promega, Madison, WI, USA) to detect cell apoptosis, following the manufacturer’s instructions [84]. In brief, 5 µm retinal sections containing optical nerve heads were deparaffinized by xylene, followed by rehydration using descending series of ethanol. Proteinase K in PBS (1:500) was used for antigen retrieval. Retinal sections were then permeabilized with an equilibration buffer for 5 min and stained with TUNEL reagents at room temperature for 1 h. Subsequently, retinal sections were counterstained with 4′,6-diamidino-2-phenylindole (DAPI) to label the nuclei. The images were captured under the ×20 objective using a fluorescence microscope (Eclipse 80i, Nikon). The number of TUNEL-positive cells in ONL was quantified as previously described [27]. TUNEL-stained retinal sections with 900 pixels × 900 pixels were chosen for manual TUNEL-positive cell counting with the help of ImageJ. The number of TUNEL-positive cells per mm was defined as the number of TUNEL-positive cells by the retina length in the selected area. 

### 3.17. Statistical Analysis 

Data were expressed as mean ± standard error of the mean (SEM) or standard deviation (SD). The graphs and statistical analysis were generated and evaluated using GraphPad Prism 10 (GraphPad). The normality of variances was confirmed before performing parametric analysis and applying the appropriate post hoc tests. The significance level was set at 0.05. Data were compared using an unpaired *t*-test for studies with two experimental groups. One-way analysis of variance (ANOVA) followed by Bonferroni or Dunnett’s post-test was used to analyze three or more experimental groups. 

## 4. Conclusions

Treating chronic retinal eye diseases remains challenging due to the difficulties in delivering drugs to the posterior segment of the eye. ECT has the potential to deliver freshly synthesized drugs in a sustained and controlled manner. In this study, the safety, performance, termination, and efficacy of the CAC ECT gel were evaluated. The results showed that the presence of CAC ECT gels in the rabbit vitreous did not adversely affect retinal function or structure, with minimal glial activation observed due to the invasive surgery. CAC ECT gels in rabbit eyes exhibited suitable encapsulation power without cell leakage, shear-stable mechanical property, resistance to degradation, and molecule permeability. Encapsulated cells remained viable with the ability to secrete freshly synthesized GDNF into the rabbit vitreous in a controlled and sustained manner. Additionally, a biosafety switch using 0.1 mg/mL Dox, an FDA-approved antibiotic, was effective in terminating the gel without damaging retinal function and architecture. In rabbits with retinal degeneration, CAC ECT gel treatment significantly improved retinal function (increase of [14% and 19% in the retinal function response for the three-gel and six-gel groups, respectively]) and preserved photoreceptor cells by reducing apoptosis.

Taking into consideration the minimally invasive surgery procedure, lack of toxicity inside the vitreous, good biocompatibility, mechanical stability, required resistance to degradation, controlled GDNF levels, and therapeutic efficacy of the CAC ECT gel, we envisage that the CAC gel has the potential to deliver the therapeutic drug to the posterior segment of the eye for the management of RDDs in clinical application without the requirement of immunosuppressants. 

## 5. Limitations of This Study and Prospective Directions 

Our study had several limitations. Firstly, we did not investigate the chronic effect of gel implantation on retinal homeostasis over an extended period. Future studies are warranted to evaluate the long-term effects of terminated gels after Dox administration on the retina. In our current study, we demonstrated the therapeutic efficacy of the CAC ECT gel in rabbits with retinal degeneration induced by SI. However, the higher GDNF levels after six gel injections when compared to three gel injections did not yield more protection; this may be due to the retinal degeneration model used. Compared to SI-induced rabbits, transgenic Pro347 Leu rabbits exhibited a slower and more progressive degeneration of photoreceptors, which is highly similar to the clinical situation in patients with RDDs [85]. This model may help better understand the association between therapeutic outcomes and the number of CAC ECT gels. Additionally, the fabrication of CAC ECT gels is labor-intensive and time-consuming (at least 5 h) with a success rate of 30–40%, requiring highly trained technicians. Therefore, improving our fabrication skills is crucial for translating these findings into clinical use.

## Figures and Tables

**Figure 1 marinedrugs-22-00394-f001:**
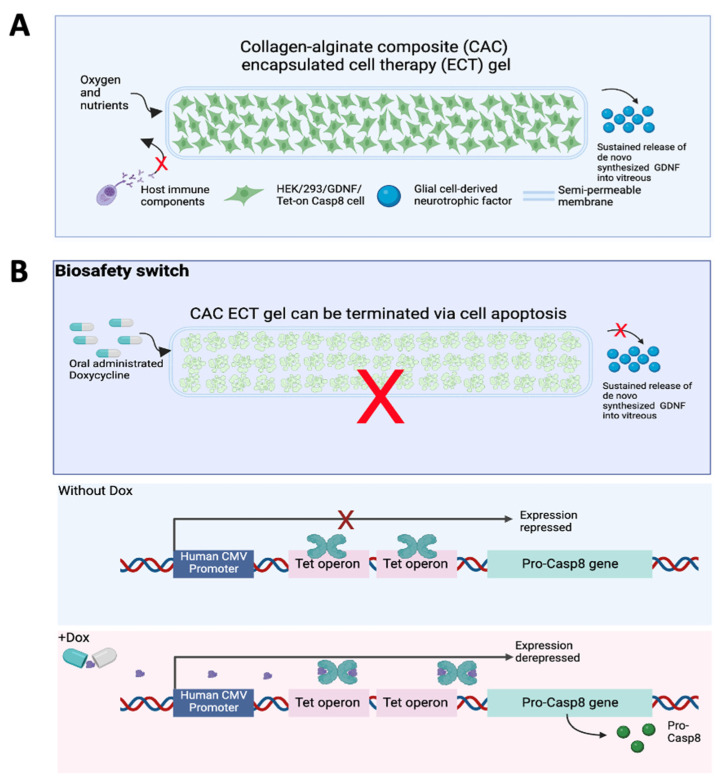
Schematic diagrams illustrating a CAC ECT gel designed for drug delivery and equipped with a controllable biosafety switch. (**A**), Gel safety and performance were assessed in healthy NZW rabbits. (**B**), Gel termination can be achieved by Dox administration when necessary and by its biosafety termination mechanism. CAC ECT, collagen–alginate composite encapsulated cell therapy. CAC ECT, collagen–alginate composite encapsulated cell therapy. NZW, New Zealand White. HEK, human embryonic kidney. GDNF, glial-cell-derived neurotrophic factor. CMV, cytomegalovirus.

**Figure 2 marinedrugs-22-00394-f002:**
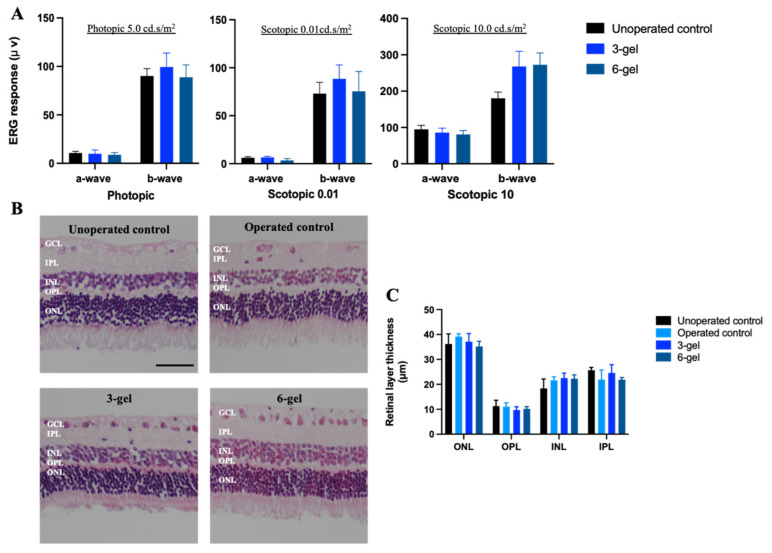
Intravitreal injection of CAC ECT gels induced no changes in retinal function and retinal cytoarchitecture (**A**). The photopic and scotopic electroretinographic responses indicated retinal function of different groups, including the unoperated control, three-gel, and six-gel groups at 2 weeks. Photopic 5 a-wave and b-wave (mean ± SEM, n = 11, 4, and 3 animals for unoperated control, three-gel, and six-gel groups, respectively), scotopic 0.01 a-wave and b-wave (mean ± SEM, n = 11, 4, and 6 animals for unoperated control, three-gel, and six-gel groups, respectively), and scotopic 10 a-wave and b-wave (mean ± SEM, n = 12, 6, and 4 animals for unoperated control, three-gel, and six-gel groups, respectively). (**B**) Representative images of H&E staining in rabbit retina: unoperated control, operated control, three-gel, and six-gel. Scale bar, 50 μm. (**C**) The thickness of ONL, OPL, INL, and IPL in rabbits sacrificed at 2 weeks post operation. No significant changes in retinal thickness were obtained after one-way ANOVA followed by Bonferroni’s post hoc comparisons tests. (mean ± SEM, n = 5, 3, 5, and 5 animals for unoperated control, operated control, three-gels, and six-gel, respectively). CAC ECT, collagen–alginate composite encapsulated cell therapy. ONL, outer nuclear layer. OPL, outer plexiform layer. INL, inner nuclear layer. IPL, inner plexiform layer. GCL, ganglion cell layer.

**Figure 3 marinedrugs-22-00394-f003:**
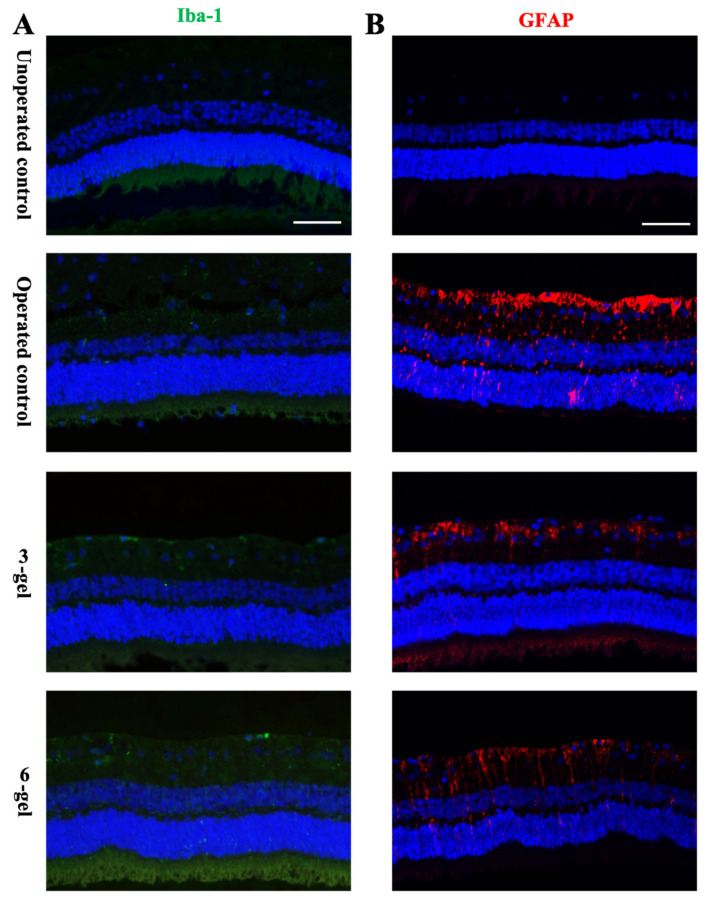
CAC ECT gel may cause stress in the retina, as assessed by gliosis in Müller cells. (**A**,**B**) CAC ECT gels were injected into the vitreous cavity, and rabbits were euthanized 2 weeks later. Immunostaining of retinal paraffin was performed to detect microglial cells (Iba-1) (**A**) and astrocytes (GFAP) (**B**), with nuclear staining by DAPI (blue). Scale bar, 50 µm. CAC ECT, collagen–alginate composite encapsulated cell therapy.

**Figure 4 marinedrugs-22-00394-f004:**
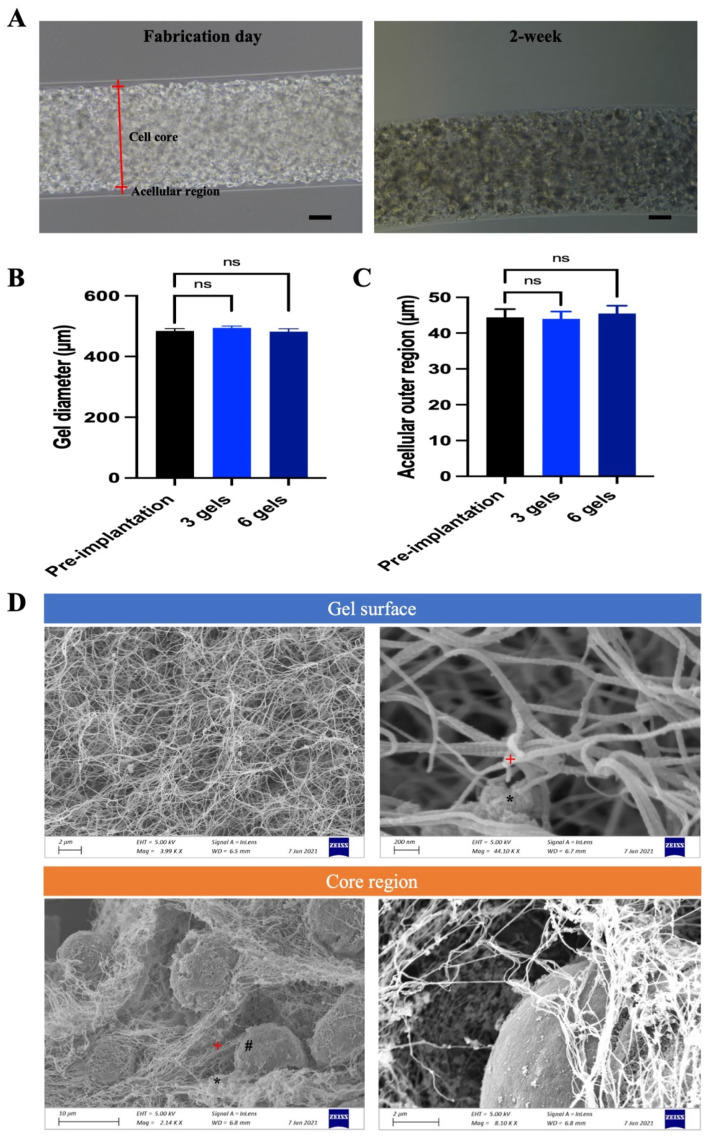
In vivo performance of CAC ECT gel after 2-week implantation. (**A**) Morphology of gels on fabrication day and retrieved gels after 2-week implantation. Scale bar, 100 μm. (**B**) Diameter of gel before and after implantation (mean ± SEM, n = 6, 6, and 6 for pre-implantation level, three-gel, and six-gel groups, respectively); one-way ANOVA with Dunnett’s post hoc test demonstrated no significant difference. (**C**) Diameter of acellular outer region (mean ± SEM, n = 6, 6, and 6 for pre-implantation level, three-gel, and six-gel groups, respectively). One-way ANOVA with Dunnett’s post hoc test demonstrated no significant difference. (**D**) SEM images showing the microstructure of the CAC ECT gel, focusing on gel surface and core region after 2-week implantation. Top row: A porous IPN network consisting of collagen and alginate assembled on the surface of the CAC gel and a magnified view of the porous network. Bottom row: cells were surrounded by collagen fibrils and CAC matrix and a magnified view of the internal structure (*: alginate, +: collagen, #: cell). CAC, collagen–alginate composite. SEM, scanning electron microscopy. IPN, interpenetrating network. ns, not significant.

**Figure 5 marinedrugs-22-00394-f005:**
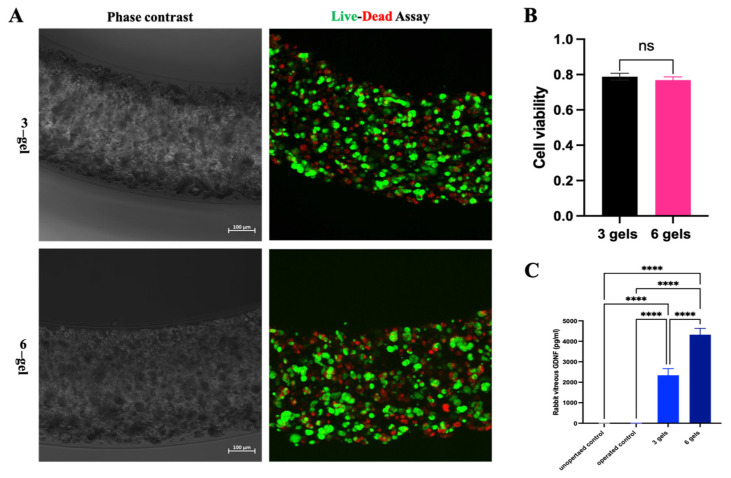
Retrieved gels were mostly viable at 2 weeks and secreted GDNF in a dose-dependent manner. (**A**) Morphology (left column) and Live/Dead image (right column) of retrieved CAC ECT gels from the three-gel and six-gel groups after 2 weeks of implantation. Scale bar, 100 μm. (**B**) Viability of retrieved gels from the three-gel and six-gel groups at 2 weeks assessed by the Live/Dead assay (mean ± SD, n = 6, and 6 for the three-gel and six-gel groups, respectively). Unpaired *t*-test demonstrated no significant differences. ns, not significant. (**C**) Vitreous GDNF level measured by ELISA assay (mean ± SEM, n = 5, 3, 5, and 5 for unoperated control, operated control, three-gel group, and six-gel group, respectively). **** *p* < 0.0001 by one-way ANOVA followed by Bonferroni’s post hoc comparisons tests. GDNF, glial-cell-derived neurotrophic factor. ANOVA, analysis of variance. ELISA, enzyme-linked immunosorbent assay.

**Figure 6 marinedrugs-22-00394-f006:**
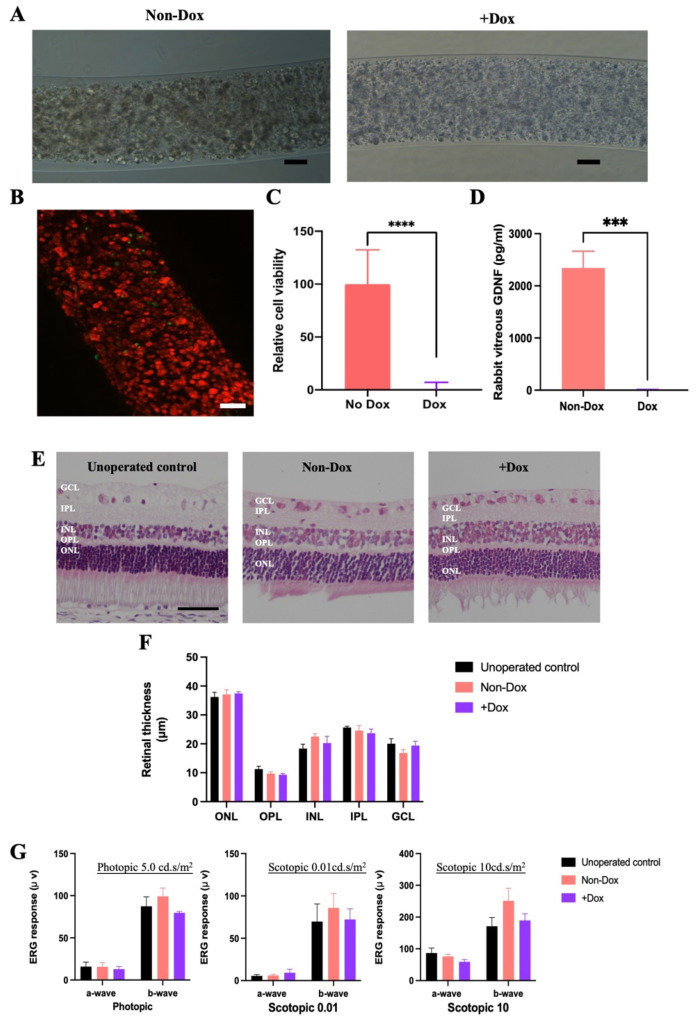
One week of Dox administration can decrease encapsulated cell viability and terminate gel functionality, without affecting retinal morphology and causing changes in retinal function. (**A**) Phase-contrast microscopic images of non-Dox-treated gels and Dox-treated gel. Scale bar, 100 μm. (**B**) Live/Dead image of retrieved gel from the Dox-treated group showing the distribution of viable cells and dying cells. Scale bar, 100 μm. (**C**) Relative cell viability of non-Dox-treated group and Dox-treated group assessed by MTS assay. (mean ± SD, n = 11, and 6 for Non-Dox and + Dox group). **** *p* < 0.0001 by unpaired *t*-test. (**D**) Accumulated GDNF levels in rabbit vitreous by GDNF ELISA after 1 week of oral Dox administration. (mean ± SEM, n = 5, and 4 for Non-Dox and +Dox group, respectively). *** *p* < 0.001 unpaired *t*-test. (**E**) H&E-stained retinal sections demonstrated no major change in retinal cytoarchitecture. Scale bar, 50 μm. (**F**) The thickness of various retinal layers including ONL, OPL, INL, IPL, and GCL was measured. (mean ± SEM, n = 5, 4 and 4 for Unoperated control, Non-Dox, and +Dox, respectively). A one-way ANOVA with Bonferroni post hoc test demonstrated non-significant differences. (**G**) The ERG response indicating the retinal functions of different groups, including Unoperated control, Non-Dox and +Dox. Amplitudes of major wave components quantified included photopic 5 a- and b-wave (mean ± SEM, n = 6, 6 and 4 animals for Unoperated control, non-Dox, and +Dox, respectively), scotopic 0.01 a-wave and b-wave (mean ± SEM, n = 6, 6 and 4 animals for Unoperated control, non-Dox, and +Dox, respectively) and scotopic 10 a-wave and b-wave (mean ± SEM, n = 6, 6 and 4 animals for Unoperated control, non-Dox, and +Dox, respectively). Statistical analysis using one-way ANOVA with Bonferroni post hoc test demonstrated no significant differences. Dox, doxycycline. GDNF, glial-cell-derived neurotrophic factor. SEM, standard deviation of the mean. MTS, dimethylthiazol-carboxymethoxyphenyl-sulfophenyl-tetrazolium. ONL, outer nuclear layer. OPL, outer plexiform layer. INL, inner nuclear layer. IPL, inner plexiform layer. GCL, ganglion cell layer.

**Figure 7 marinedrugs-22-00394-f007:**
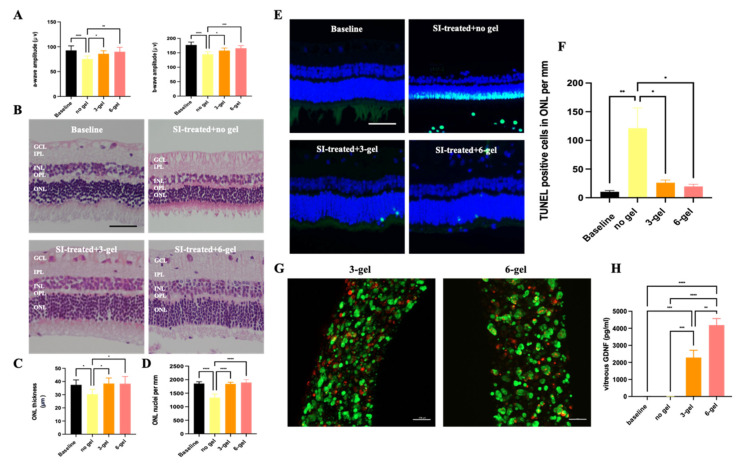
The therapeutic efficacy of CAC ECT gels in rabbits with retinal degeneration induced by SI. (**A**) Scotopic a-wave and b-wave electroretinographic responses were measured at 10 cd.s/m^2^, indicating retinal functions in normal rabbit eyes (baseline) and SI-induced rabbit eyes, including no-gel-injected eyes, and eyes injected with 3 and six gels. (mean ± SEM, n = 11, 8, 7, and 6 independent animals for baseline, no gel, three-gel, and six-gel groups). * *p* < 0.05, ** *p* < 0.01, *** *p* < 0.001, **** *p* < 0.0001 by one-way ANOVA followed by Dunnett’s post-test. (**B**) Representative image of H&E-stained retinal sections from healthy rabbits (baseline), rabbits receiving SI only, and rabbits with three-gel and six-gel treatment. Scale bar, 50 µm. (**C**) The thickness of ONL containing the photoreceptor nuclei was significantly higher in rabbits treated with CAC ECT gels when compared with no-gel-treated rabbits after 2-week implantation. (mean ± SEM, n = 6, 8, 7, and 6 for baseline, no gel, three-gel, and six-gel groups). * *p* < 0.05 by one-way ANOVA followed by Dunnett post *t*-test. (**D**) The number of ONL nuclei per mm was significantly higher in rabbits with three-gel and six-gel injection compared with no-gel-treated rabbits after 2-week implantation. (mean ± SEM, n = 6, 8, 7, and 6 for baseline, no gel, three-gel, and six-gel group). **** *p* < 0.0001 by one-way ANOVA followed by Dunnett post *t*-test. (**E**) Representative images showing the distribution of apoptotic cells in the retina detected by TUNEL assay and nuclei were stained by DAPI. Scale bar, 50 µm. (**F**) The number of TUNEL-positive apoptotic cells in ONL significantly decreased in SI-induced rabbits treated with three-gel and six-gel compared with no-gel-treated rabbits after 2-week implantation. * *p* < 0.05, ** *p* < 0.01 by one-way ANOVA followed by Dunnett’s post-test. (**G**) Live/Dead assay showed that the six-gel-treated and six-gel-treated animals contained mostly living cells (green). Scale bar, 100 μm. (**H**) Increased vitreous GDNF levels only detected in rabbits with gel treatment. ** *p* < 0.01; *** *p* < 0.001; **** *p* < 0.0001 by one-way ANOVA with Bonferroni post hoc. CAC ECT, collagen–alginate composite encapsulated cell therapy. SI, sodium iodate. ANOVA, analysis of variance. ONL, outer nuclear layer. GDNF, glial-cell-derived neurotrophic factor. TUNEL, terminal deoxynucleotidyl transferase dUTP nick end labeling. DAPI, 4′,6-diamidino-2-phenylindole.

**Figure 8 marinedrugs-22-00394-f008:**
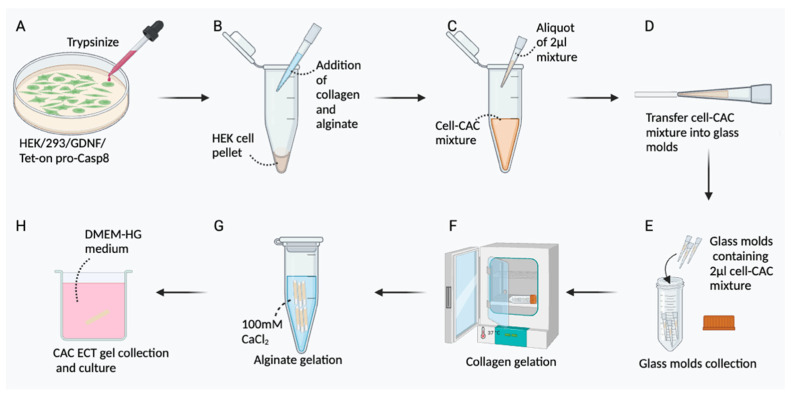
Schematic diagram showing the detailed preparation of CAC ECT gels. (**A**) HEK/293/GDNF/Tet-on pro-Casp8 cells were trypsinized. (**B**) The HEK/293/GDNF/Tet-on pro-Casp8 cell pellet was mixed with collagen and alginate. (**C**) An aliquot of 2 μL of the gel mixture was prepared. (**D**) The 2 μL gel mixture was gently transferred to the glass molds. (**E**) The glass molds containing the 2 μL gel were collected into a 50 mL Falcon tube. (**F**) The glass molds were then placed in an incubator at 37 °C for 75 min to initiate collagen gelation. (**G**) The glass molds were then introduced to a 100 mM CaCl_2_ solution followed by a CaCl_2_ bath to gelate the alginate. (**H**) The gels were collected from the molds and cultured in a DMEM-HG medium before intravitreal gel injection.

**Figure 9 marinedrugs-22-00394-f009:**
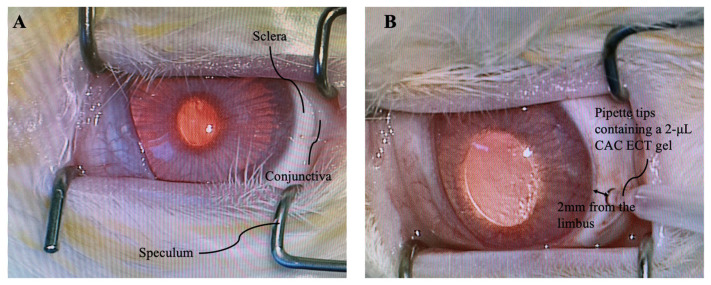
Intravitreal injection of CAC ECT gels on healthy NZW rabbits. (**A**) The surrounding ocular tissue was first sterilized by diluted betadine. Then, eyelids were spread apart using a speculum. (**B**) One incision (2 mm from the limbus) into the vitreous was made using a 20-gauge blade followed by an insertion of a pipette tip containing a 2-μL CAC ECT gel into the vitreous.

**Table 1 marinedrugs-22-00394-t001:** The CAC ECT gel did not cause any alterations in the body weight of the animals and induced a short-term decrease in intraocular pressure (IOP). The body weight and IOP of the rabbits were assessed post gel implantation after 2 weeks. IOP was measured bilaterally in animals after operated control or gel implantation procedure at 2 weeks. CAC ECT, collagen–alginate composite encapsulated cell therapy. IOP, intraocular pressure.

	Unoperated Control	Operated Control	Three-Gel	Six-Gel
Body weight (kg)	3.23 ± 0.17	3.35 ± 0.12	3.27 ± 0.22	3.33 ± 0.19
IOP (mmHg)	18.75 ± 0.77	17.13 ± 0.74	14.27 ± 0.74	13.5 ± 0.70

## Data Availability

The data that support the findings of this study are provided in this paper and Appendix A.

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
