# Peer review of "Bioactive Glial-Derived Neurotrophic Factor from a Safe Injectable Collagen–Alginate Composite Gel Rescues Retinal Photoreceptors from Retinal Degeneration in Rabbits"

_marinedrugs, 2024, doi:10.3390/md22090394_

Round 1

Reviewer 1 Report

Comments and Suggestions for Authors

The manuscript entitled “Bioactive Glial-Derived Neurotrophic Factor from a Safe Injectable Collagen-Alginate Composite Gel Rescues Retinal Photoreceptors from Retinal degeneration in Rabbits” explored the clinical potential of therapeutic efficacy of CAC ECT gel with sodium iodate (SI)-induced retinal degeneration in rabbits. Results of the study demonstrate that the gel can be safely implanted without causing damage to the retina or lens, resists degradation, promotes cell attachment, and secretes bioactive GDNF. The GDNF levels could be adjusted based on the number of implants. In addition, Dox effectively terminated the gel's function without causing any damage to the retina. The gel significantly improved a-wave and b-wave amplitudes and reduced photoreceptor apoptosis in rabbits with retinal degeneration. From these findings, the authors concluded that CAC ECT gel could potentially eliminate the need for immunosuppressive medications to treat various retinal diseases. The current work is well-designed, and the present findings are interesting. 

Comments:     

1) In the introduction section, the authors are advised to elaborate on the experimental model (Sodium iodate (SI)-induced retinal degeneration in rabbits), its advantages, and how it mimics the human condition.

2) lines 88-90, the authors state “Our earlier studies demonstrated that CAC ECT gel had shown promising therapeutic potential in rescuing photoreceptor cells in dystrophic Royal College of Surgeons rats, a rat model of photoreceptor degeneration”. This statement is currently unsupported by references. The authors are encouraged to include appropriate citations to substantiate this claim.

3) In section 3.4., the authors are recommended to clearly describe the experimental design, including the number of groups, the number of animals per group, and the rationale behind the selection of these numbers, with appropriate references.

4) To enhance clarity for readers, the design of the current animal study should be illustrated in a separate figure.

5) In the material and methods section, please comment on the adequacy of sample size calculation. How did the authors decide on using the listed number of animals per experimental group? Did you use power analysis? I would suggest that authors address this point and add the answers/proper citations to the material and methods section.

6)  Include a statement on randomization and blinding. No problem if the experiments were not randomized/blinded, just state within your manuscript.

7) In the entire Material and methods section, the authors are advised to add the full source of reagents and kits. For example, in lines 632-633, the authors state “Rabbits were anaesthetized by an intramuscular injection of a mixture of xylazine (5 mg/kg) and ketamine (45 mg/kg) (Alfasan lab)”. Please, specify the country, state, and city.

8) The authors are advised to add the cat no. for the used chemicals and antibodies.

9) In section 3.11, how did the authors decide on using 3-6 units of CAC ECT gel? how is the dose of in rabbits relevant for human translation? Can you discuss the dose used for possible translation in humans, for example, by using conversion tables available in the literature using the Human effective dose (HED) formula= animal dose x animal Km/ human Km (Nair AB, Jacob S. A simple practice guide for dose conversion between animals and humans. J Basic Clin Pharm. 2016 Mar;7(2):27-31). Authors are advised to address this point and add the answers/proper citations to this section.

10) The authors are advised to add the cat no. for the used kits and antibodies.

11) In section 3.8. (Immunohistostaining), the authors should clarify whether a negative control was performed in these experiments to ensure specific binding of the antibody to the target protein. This information should be added to this section.

12) In the statistical analysis section, did the authors check data normality before proceeding to one-way ANOVA? Authors are advised to address this point and add the answers in this section.

13) To make all figure legends stand-alone, authors are advised to add the full name of the used abbreviations at the end of each legend.

14) Quantification of Iba-1 immunostaining is required for the data generated in Figure 3.

15) In Figure 5, quantification of panel A is required. Likewise, the same is required in Figures 6B and 7G.

Author Response

Reviewer 1

Comments 1:   In the introduction section, the authors are advised to elaborate on the experimental model (Sodium iodate (SI)-induced retinal degeneration in rabbits), its advantages, and how it mimics the human condition.

Response to the comments 1: Thank you for your valuable feedback. We have taken your suggestion into consideration and have elaborated on the experimental model used in our study, specifically the Sodium iodate (SI)-induced retinal degeneration. In the introduction section, we have now detailed the advantages of this model and explained how it closely mimics the human condition of retinal degeneration. The added information is as follows, which can be found in line 104-109.

“Sodium iodate is a small molecule that has been widely used for inducing retinal degeneration. It selectively damages retinal pigment epithelium (RPE) and subsequent photoreceptor cell death through oxidative stress. The sodium iodate-induced retinal degeneration model is advantageous because it reliably mimics dry AMD. Additionally, it offers rapid and reproducible onset of damage, making it a versatile and cost-effective tool for testing potential drug treatments”.

We believe this additional information will provide a clearer understanding of our experimental approach and its relevance to human disease.

Comments 2: lines 88-90, the authors state “Our earlier studies demonstrated that CAC ECT gel had shown promising therapeutic potential in rescuing photoreceptor cells in dystrophic Royal College of Surgeons rats, a rat model of photoreceptor degeneration”. This statement is currently unsupported by references. The authors are encouraged to include appropriate citations to substantiate this claim.

Response to the comments 2:Many thanks for your comment. We have included the citation. Please see line 93, reference 28.

Comments 3: In section 3.4., the authors are recommended to clearly describe the experimental design, including the number of groups, the number of animals per group, and the rationale behind the selection of these numbers, with appropriate references.

Response to the comments 3: Thank you for your comment. We would like to clarify that Section 3.4 specifically describes the intravitreal procedure of CAC ECT gel. Therefore, we believe it is not the appropriate section to detail the experimental design. The experimental design is thoroughly described in Sections 3.9, 3.10, 3.11, and 3.12. Additionally, in line with the 3R principle (Replacement, Reduction and Refinement), each rabbit group includes at least three animals. Please refer to the mentioned sections for further details. Please see lines 812-825, 827-832, 838-839, 846-852. Moreover, in the original manuscript, we indicated the n number for each group of rabbits in every figure legend. The added information is as follows.

3.9 in vivo gel safety study

In this study, NZW rabbits were randomly assigned into3 groups, each consisting of at least 3 animals for the in vivo gel safety study. The groups were designated as follows: Group 1 served operated control and received intravitreal gel procedure without gel treatment; Group 2 received intravitreal injection of three units of gel and Group 3 received intravitreal injection of six units of gel. Following the administration of intravitreal gel procedure and either three or six units of gel into the eyes, the rabbits were monitored over a 2-week period. Retinal detachment and cataract formation were assessed using a binocular ophthalmoscope to evaluate the structural integrity of the eye. Retinal function was measured via ERG to detect any functional impairments. After the 2-week observation period, the rabbits were sacrificed, and their eyes were collected for further analysis. Retinal stress markers were evaluated using immunohistochemistry (IHC), while retinal morphology was assessed through hematoxylin and eosin (HE) staining to examine any histological changes (Fig. 10A).

3.10 In vivo gel performance study

In this study, healthy NZW rabbits were randomly divided into four groups of at least three animals each for the in vivo gel performance study. Group 1 served as the unoperated control without any treatment; Group 2 served as operated control received intravitreal gel procedure without gel treatment. Group 3 served as 3-gel group received intravitreal injection of three units of gel and Group 4 received intravitreal injection of six units of gel. The rabbits were sacrificed 2 weeks post-injection.

3.11 In vivo gel termination study

In this study, healthy NZW rabbits were randomly divided into two groups of at least four animals each for the in vivo gel termination study.

3.12 in vivo therapeutic potential of CAC ECT gel

In this study, rabbits received with intravenous injection of sodium iodate (SI) were randomly divided into three groups of at least six animals each for the in vivo therapeutic potential study. Group 1 served as no gel control received intravenous SI injection without gel treatment. Group 2 served as 3-gel group received intravenous SI injection and intravitreal injection of three units of gel and Group 3 received intravenous SI injection and intravitreal injection of six units of gel. healthy NZW rabbits without any treatment served as baseline.”

Comments 4: To enhance clarity for readers, the design of the current animal study should be illustrated in a separate figure.

Response to comments 4: Thank you for your comment. We have created a separate figure (Figure 10) to clearly illustrate the design of the current animal study, distinct from the concept of CAC ECT gel (Figure 1). Additionally, we have added a section in the "Materials and Methods" to describe the in vivo gel safety study in more details. Please refer to section 3.9 for more information, which can be found in lines 812-825. The added information is as follows.

“3.9 in vivo gel safety study

In this study, New Zealand White (NZW) rabbits were randomly assigned into3 groups, each consisting of at least 3 animals for the in vivo gel safety study. The groups were designated as follows: Group 1 served operated control and received intravitreal gel procedure without gel treatment; Group 2 received intravitreal injection of three units of gel and Group 3 received intravitreal injection of six units of gel. Following the administration of intravitreal gel procedure and either three or six units of gel into the eyes, the rabbits were monitored over a 2-week period. Retinal detachment and cataract formation were assessed using a binocular ophthalmoscope to evaluate the structural integrity of the eye. Retinal function was measured via ERG to detect any functional impairments. After the 2-week observation period, the rabbits were sacrificed, and their eyes were collected for further analysis. Retinal stress markers were evaluated using immunohistochemistry (IHC), while retinal morphology was assessed through hematoxylin and eosin (HE) staining to examine any histological changes (Fig. 10A).”

Comments 5: In the material and methods section, please comment on the adequacy of sample size calculation. How did the authors decide on using the listed number of animals per experimental group? Did you use power analysis? I would suggest that authors address this point and add the answers/proper citations to the material and methods section.

Response to the comments5:Thank you for your valuable suggestion regarding the sample size calculation. Since this is a proof-of-concept study, we did not perform a power analysis for our study due to the following reasons: 1) Previous Experience: Based on our laboratory's prior studies, we have established that the chosen number of rabbits is sufficient to detect significant biological effects. Please refer to our previous publication for information [1]. The purpose of this experiment is to demonstrate whether CAC ECT Gel can serve as an intraocular drug delivery platform for treating retinal diseases. Typically, biological experiments require at least three biological replicates (n=3) for each experimental parameter. To meet this requirement, we used at least three rabbits (n=3) per experimental group to assess the intraocular safety, performance, termination capability, and therapeutic potential of the gel. 2) Resource and Ethical Considerations: Given the high cost of using large animals and the associated ethical considerations, we ensured that each group included no fewer than 3 rabbits to minimize animal use while maintaining the validity of the experiment.

Comments 6:  Include a statement on randomization and blinding. No problem if the experiments were not randomized/blinded, just state within your manuscript.

Response to the comment 6: Thank you for your suggestion. We acknowledge the importance of randomization and blinding in reducing bias in experimental studies. In our study, the rabbit experiments were randomized. We have included a clear statement that rabbits were randomly divided into different groups in section 3.9. 3.10, 3.11 and 3.12 in the manuscript to indicate this. We have added a Section 3. 3 to state the experiments were carried out blindly, which can be found in lines 722-725. The added information is as follows.

“All researchers involved in the assessment of outcomes, including those performing the procedures, monitoring, and analyzing the results, were blinded to the treatment groups to minimize bias. The code for group assignment was only broken after the completion of data analysis.”

Comments 7: In the entire Material and methods section, the authors are advised to add the full source of reagents and kits. For example, in lines 632-633, the authors state “Rabbits were anaesthetized by an intramuscular injection of a mixture of xylazine (5 mg/kg) and ketamine (45 mg/kg) (Alfasan lab)”. Please, specify the country, state, and city.

Response to the comments 7: Thank you for your suggestion. We have revised the Materials and Methods section to include the full source information for all reagents and kits used. Please see the updated details in the revised manuscript. please see Line 728.

Comments 8: The authors are advised to add the cat no. for the used chemicals and antibodies.

Response to comment 8: Many thanks for your comment. We have added the cat no. for the used chemicals and antibodies. Please see updated details in the revised manuscript. Please see lines 799-805. The added information is as follows.

“anti-glial fibrillary acidic protein (GFAP) antibody (1:500; Cat# 14-9892-82, ThermoFisher, Waltham, Massachusetts, USA) and anti-ionized calcium-binding adaptor molecule 1 (Iba-1) antibody (1:500, catalog # 019-19741, FUJIFILM Wako Pure Chemical Corporation, Japan) overnight at 4°C, and later with Alexa Fluor 568 goat anti-mouse IgG (Invitrogen,Cat#A-11004, Waltham, Massachusetts, USA) and Alexa Fluor 488 goat anti-rabbit IgG (Invitrogen, Cat# A-11008, Waltham, Massachusetts, USA) for 1 hour at room temperature”

Comments 9: In section 3.11, how did the authors decide on using 3-6 units of CAC ECT gel? how is the dose of in rabbits relevant for human translation? Can you discuss the dose used for possible translation in humans, for example, by using conversion tables available in the literature using the Human effective dose (HED) formula= animal dose x animal Km/ human Km (Nair AB, Jacob S. A simple practice guide for dose conversion between animals and humans. J Basic Clin Pharm. 2016 Mar;7(2):27-31). Authors are advised to address this point and add the answers/proper citations to this section.

Response to the comments: Thank you for your insightful comments. We chose to use 3 and 6 units of CAC ECT gel based on guidelines from the American Academy of Ophthalmology, which suggest that an injection volume of 0.05 mL is commonly used for intravitreal injections (reference: https://www.aao.org/eyenet/article/how-to-give-intravitreal-injections). Each unit of our CAC ECT gel is 2 µL, as detailed in Section 3.2. To ensure successful injection into the vitreous, we also used a small amount of sterile medium. Thus, each injection volume, including the gel and medium, was controlled to approximately 10 µL. Therefore, we chose a maximum of 6 units of gel to accommodate this volume while maintaining procedural consistency. The use of 3 units of gel was intended to investigate whether the release of GNDF correlates linearly with the number of CAC ECT units injected. We have carefully considered your suggestion regarding the use of the Human Effective Dose (HED) formula for translating animal doses to human equivalents. However, we believe that Section 3.11 in original manuscript, which focuses primarily on the in vivo gel therapeutic efficacy experimental design, may not be the most appropriate place to include such a discussion. Nonetheless, we acknowledge the importance of dose translation and have addressed this topic in the Results and Discussion section. You can find this discussion on lines 668-670 of the manuscript. We have included proper citations (please refer to reference 80). We hope this adequately addresses your concern. The added information is as follows.

“Although there are currently formulas available to convert animal doses to human equivalent doses and vice versa [2], their applicability in our study requires further investigation when considering our study focuses on intravitreal administration. ”

Comments 10: The authors are advised to add the cat no. for the used kits and antibodies.

Response to the comments 10: Many thanks for your comment. We have added the cat no. for the used kits and antibodies. Please see updated revised manuscript.

Comments 11: In section 3.8. (Immunohistostaining), the authors should clarify whether a negative control was performed in these experiments to ensure specific binding of the antibody to the target protein. This information should be added to this section.

Response to the comments: Many thanks for your comment. In the original manuscript, we have already clarified that a negative control was included to ensure specific binding, which the reviewer can find in lines 805-806. The information is as follows.

“Negative controls involved substituting the primary antibody with a blocking buffer (10% goat serum).”

Comments 12: In the statistical analysis section, did the authors check data normality before proceeding to one-way ANOVA? Authors are advised to address this point and add the answers in this section.

Response to the moment 12: Thank you for your insightful comment. We confirm that data normality was assessed prior to performing the one-way ANOVA. We have added this information to the Statistical Analysis section in the revised manuscript, which can be found in lines 925-926. The added information is as follows.

“The normality of variances was confirmed before performing parametric analysis and applying the appropriate post-hoc tests.”

Comments 13: To make all figure legends stand-alone, authors are advised to add the full name of the used abbreviations at the end of each legend.

Response to the comments 13: Many thanks for your comment. We have added the full name of the used abbreviation at the end of each figure legend.

Comments 14: Quantification of Iba-1 immunostaining is required for the data generated in Figure 3.

Response to the comments 14: Many thanks for your comment, we have quantified Iba-1 immunoreactivity, which can be found in supplementary data figure 1.

Comments 15: In Figure 5, quantification of panel A is required. Likewise, the same is required in Figures 6B and 7G.

Response to the comments 15: Thank you for your feedback. We would like to clarify that Figure 5B represents the quantification of panel A. Additionally, the quantifications for Figures 6B and 7G have already been completed and can be found in the supplementary figure 3 and figure 4.

Reference: 

Nair, A. B. and S. Jacob (2016). "A simple practice guide for dose conversion between animals and human." J Basic Clin Pharm 7(2): 27-31.

Williams, R., V. Kearns, A. Lo, M. Day, M. Garvey, Y. Krishna, D. Ma, T. Stappler and D. Wong (2013). "Novel Heavy Tamponade for Vitreoretinal Surgery." Investigative ophthalmology & visual science 54.

Reviewer 2 Report

Comments and Suggestions for Authors

General comments

The submitted manuscript is focused on the development of an injectable collagen-alginate composite gel, proposed as a safe intraocular drug delivery platform for the treatment of retinal degenerative diseases.

Some minor revisions have to be applied.

More details and specific remarks and suggestions are reported below point by point.

Keywords

The chosen keywords (i.e. Composite hydrogel; encapsulated cell therapy; neuroprotection; sustained drug delivery; retina) do not completely cover the manuscript content. Specific ones could be added, such as those related to the used materials, cells and so on.

1. Introduction

- The Introduction section is well organised and well-conceived.

-        The alginate has been used in numerous experimental works and more should be cited, including “Neuro-differentiated Ntera2 cancer stem cells encapsulated in alginate beads: First evidence of biological functionality. Materials Science and Engineering: C81(2017), 32-38.”

-        It is suggested to report a brief list of the used characterisations at the end of the Introduction section.

3. Materials and Methods

More details about the used reagent and solvents, such as the purity, have to be added.

3.14. Scanning Electron Microscope (SEM)

Were the samples gold coated? Which conditions were applied?

4. Conclusions

This section ha sto be made more concise and immediate. The main numerical results have to be specified. 

Comments on the Quality of English Language

The English is good.

Author Response

Comments 1: The chosen keywords (i.e. Composite hydrogel; encapsulated cell therapy; neuroprotection; sustained drug delivery; retina) do not completely cover the manuscript content. Specific ones could be added, such as those related to the used materials, cells and so on.

Response to the comments 1: Thank you for your comment. We have added more keywords. The new list is as follows.

collagen-alginate composite hydrogel, doxycycline, electroretinography, encapsulated cell therapy, HEK cells, intravitreal, neuroprotection, retina, sustained drug delivery

Comments 2: - The Introduction section is well organised and well-conceived.

-  The alginate has been used in numerous experimental works and more should be cited, including “Neuro-differentiated Ntera2 cancer stem cells encapsulated in alginate beads: First evidence of biological functionality. Materials Science and Engineering: C, 81(2017), 32-38.”

Response to the comments 2: Thank you for the suggestion. We agree that citing additional studies on the use of alginate in experimental work would strengthen our manuscript. We have now included the recommended citation, along with other relevant references, in the revised manuscript. Please see line 84, reference 23.

Comments 3: It is suggested to report a brief list of the used characterisations at the end of the Introduction section.

Response to the comments 3: We agree to provide the gel characterizations. However, to maintain the conciseness of the main text, we have included these in the supplementary table1.  

Comments 4: More details about the used reagent and solvents, such as the purity, have to be added.

Response to the comments 4: We have added more detailed information about the reagents and solvents used, including their purity.

Comments 5: 3.14. Scanning Electron Microscope (SEM)

Were the samples gold coated? Which conditions were applied?

Response to the comments 5: Thank you for your valuable feedback. Yes, the samples were gold-coated prior to analysis. In original manuscript, we have clearly stated the condition please see lines 904-9065. The information is as follows.

“samples were then sputtered with gold (SCD 005, Bal-tec).”

  1. Conclusions

Comments 6: This section has to be made more concise and immediate. The main numerical results have to be specified.

Response to the comments 6: Thank you for your comments. we have revised our conclusion to make it sound more concise and immediate, which can be found in lines 940-945. Beside it, we also included numerical results in result section. Please refer to lines 193-199,287-297, 513-525, 568-577, 579-590, and 593-598.

The information is as follows:

“Specifically, the ONL thickness was 37.10±1.64 µm in the 3-gel group, 35.19±1.04µm in the 6-gel group, and 39.16±0.64µm in the operated control. The OPL thickness was 9.69±0.67µm in the 3-gel group, 10.20±0.42µm in the 6-gel group, and 11.08±0.87µm in the operated control. Similarly, the INL thickness was 22.50±1.01µm in the 3-gel group, 22.29±0.76µm in the 6-gel group, and 21.62±0.81µm in the operated control, while the IPL thickness was 24.61±1.64µm in the 3-gel group, 21.87±0.45µm in the 6-gel group, and 21.89±2.22µm in the operated control.

Indeed, retrieved gels in the 3-gel and 6-gel groups showed no significant changes in gel diameter when compared to the pre-implantation level. The average gel diameter was 494.20±6.12µm in the 3-gel group and 481.70±9.42µm in the 6-gel group, which were not significantly different from the pre-implantation diameter (484.42±7.44µm) (Fig. 4B). This highlights that the CAC system exhibited good resistance to degradation over 2 weeks. Analysis of the thickness of the acellular outer region before and after implantation demonstrated no significant difference, with the thickness measured at 44.40±2.28µm before implantation and 43.97±2.05µm in the 3-gel and 45.49±2.14µm after implantation (Fig. 4C). This is consistent with previous morphological findings that no cell protrusion was spotted after 2-week implantation.

Analysis of the thickness of various retinal layers revealed no significant differences among the unoperated control, non-Dox treated group, and Dox treated group (Fig. 6F). Specifically, the thickness of the ONL was 36.24±1.60µm in the unoperated control, 37.10±1.64µm in the non-Dox treated group, and 37.43±0.59µm in the Dox treated group. The OPL thickness was 11.26±0.96µm in the unoperated control, 9.69±0.67 µm in the non-Dox treated group, and 9.33±0.43µm in the Dox treated group. Similarly, the INL thickness was 18.40±1.55µm in the unoperated control, 22.50±1.01µm in the non-Dox treated group, and 20.29±2.30µm in the Dox treated group. The IPL thickness was 25.63±0.45µm in the unoperated control, 24.61±1.64 µm in the non-Dox treated group, and 23.67±1.44µm in the Dox treated group. The GCL thickness was 20.05±1.73µm in the unoperated control, 16.80±1.22µm in the non-Dox treated group, and 19.40±1.49µm in the Dox treated group. These results suggest that oral Dox administration imposed no detrimental damage to retinal architecture.

The amplitude of the a-wave in SI-treated rabbits without gel injection was significantly lower than the baseline level, with measured value of 75.35±6.04µV compared to the baseline level of 92.62±9.16µV (Fig. 7A). This reduction was alleviated by 3-gel and 6-gel treatment, with a-wave amplitudes of 85.88±6.08µV and 89.99±8.88µV, respectively, for the 3-gel and 6-gel groups. Similarly, both the 3-gel and 6-gel groups exhibited significant recovery in scotopic b-wave amplitude compared to the SI-only group, which had a b-wave amplitude of 144.78±8.85µV. The 3-gel group showed a b-wave amplitude of157.78±8.58µV, while the 6-gel group demonstrated an even greater recovery with a b-wave amplitude of 165.79±8.88µV.

Compared to the baseline level, rabbits that received SI injection without gel implantation exhibited a marked reduction in ONL thickness, with an average thickness of 30.37±3.75 µm compared to the baseline thickness of 37.46±3.78 µm (Fig. 7C). Similarly, the number of ONL nuclei per mm of retina was significantly reduced to 1337±134 nuclei/mm, compared to the baseline count of 1856±65 nuclei/mm (Fig. 7D). However, both 3-gel and 6-gel treatment groups displayed remarkable increases in ONL thickness, with the 3-gel group showing an average thickness of 38.49±4.20 µm and the 6-gel group showing 38.34±5.49 µm (Fig. 7C). The number of ONL nuclei per mm also increased significantly when compared to the no-gel-treated rabbits, with the 3-gel group exhibiting 1843±60 nuclei/mm and the 6-gel group reaching 1893±107 nuclei/mm (Fig. 7D). These findings suggest that CAC ECT gel was effective in rescuing photoreceptors in the rabbit model of SI-induced retinal degeneration.

A notable increase in the number of TUNEL-positive apoptotic cells in the ONL was observed in the SI-treated group that did not receive gel treatment (121±35cells per mm) compared to baseline (10±2cells per mm). However, the number of TUNEL-positive cells in the ONL significantly decreased in both the 3-gel (26± 4cells per mm) and 6-gel (20±4 per mm) treatment groups compared to non-gel treated rabbit retinas (Fig. 7F).

Reviewer 3 Report

Comments and Suggestions for Authors

The study aimed to assess the ability of injectable collagen-alginate composite gel for safe and efficient delivery of glial cell line-derived neurotrophic factor, however the should be addressed

1-The abstract section did not contain any numerical findings; please rewrite.

2- The authors mentioned that the gel was designed for sustained release of glial cell line-derived neurotrophic factor (GDNF). However, no release study was performed to assess if there was any burst  release or dose dumping. 

3-In the introduction section, the authors mentioned in line 78 their pervious study regarding collagen alginate composite encapsulated cell therapy, please cite your published study in this part with a heighlight regarding the novelty in the current study.

4- Please revise all the material including all chemicals and equipment mentioned in the manuscript by adding  the origin (city & country) of manifacturing company.

5- Section 3.4 lack references for all the anaesthesia doses and protocol procedures. same notice for section 3.6 ,3.7, 3.9, 3.10, 3.13. 3.14 cite a reference for these procedures.

6- please discuss more the reasons for the absence of  cell protrusion in the current system compared to some ECT sytems which showed uncontrolled growth.

7- please discuss more the reasons for the good mechanical stability revealed by the current system compared to pervious studies conducted on alginate- collagen composite beside the level of MMP2 and low concentrations of calcium and sodium.

8- Discuss more the reasons for the selective permeability showed by the system.

Author Response

Comments 1: The abstract section did not contain any numerical findings; please rewrite.

Response to the comments 1: Thank you for your comments. We have not included specific numerical findings in the abstract to maintain a concise and focused summary of our study's objectives, methods, and key conclusions. The abstract is intended to provide a broad overview of the research, while detailed numerical results are thoroughly discussed in the Results and Discussion sections of the manuscript. However, we have revised the results section of our manuscript to include detailed numerical results. Please refer to lines 193-199,287-297, 513-525, 568-577, 579-590, and 593-598.

Comments 2: The authors mentioned that the gel was designed for sustained release of glial cell line-derived neurotrophic factor (GDNF). However, no release study was performed to assess if there was any burst release or dose dumping. 

Response to the comments 2: In this study, we did not conduct a specific assessment to determine if there was any burst release or dose dumping of GDNF. However, we measured the vitreous GDNF levels using ELISA and observed a dose-dependent release pattern across the unoperated control, operated control, 3-gel, and 6-gel groups (Fig. 5C). This pattern suggests a linear relationship between GDNF release, the number of implants, and cell activity. Additionally, our previous studies in rat eyes demonstrated an increase in GDNF release as cells continued to proliferate, supporting the conclusion that the release of GDNF is likely to be sustained over time.

Comments 3: In the introduction section, the authors mentioned in line 78 their pervious study regarding collagen alginate composite encapsulated cell therapy, please cite your published study in this part with a highlight regarding the novelty in the current study.

Response to the comments 3: Thank you for your feedback. We have included our previous study on collagen-alginate composite encapsulated cell therapy in the introduction section (please refer to reference 28). We have already highlighted the novelty of current study in our original manuscript. The novelty of the current study lies in the translation of CAC ECT gel from laboratory into clinical application. The rodent eye has been criticized for its small eye size compared to that of humans, the anatomical difference and species difference in drug pharmacokinetics, and limited vitreous capacity. The rabbit is a commonly used animal model in ophthalmic studies due to its eye size being similar to that of humans, ease of handling, availability, and low cost. Moreover, compared to rodents, rabbit eyes have enough space to accommodate multiple ocular implants, making it possible to evaluate the multiple gel performance and therapeutic efficacy in clinical situation. This study also explores the effects of different gel concentrations on retinal disease model and functionality, providing new insights that were not addressed in our previous research.

Comments 4: Please revise all the material including all chemicals and equipment mentioned in the manuscript by adding the origin (city & country) of manufacturing company.

Response to the comments 4: Thank you for your insightful feedback. We will carefully revise the manuscript to include the origin (city and country) of the manufacturing companies for all chemicals and equipment mentioned.

Comments 5: Section 3.4 lack references for all the anaesthesia doses and protocol procedures. same notice for Sections 3.6 ,3.7, 3.9, 3.10, 3.13. 3.14 cite a reference for these procedures.

Response to the comments 5 : Thank you for your feedback. We have added references for the anesthesia dosages and protocol procedures, as well as for the intravitreal procedure in the manuscript (please refer to line 728-730, reference 82 and 28). Regarding Sections 3.9, 3.10, 3.13, and 3.14, which focus primarily on IHC staining, SEM, and the experimental design for rabbit grouping, detailed descriptions of the anesthesia dosages were not included to avoid redundancy. We have ensured that Sections 3.4 and 3.7 provide sufficient details for the anesthesia procedures and sacrifice dosage. We appreciate your understanding and hope this addresses your concerns.

Comments 6: please discuss more the reasons for the absence of cell protrusion in the current system compared to some ECT systems which showed uncontrolled growth.

Response to the comments 6: Thank you for your comments. we have discussed the absence of cell protrusion system compared to some ECT system, which can be found in lines 277-282. The added information is as follows.

“In our previous study, we found that using a 2-stage protocol to crosslink CAC ECT gel, where collagen was crosslinked first followed by alginate gelation, led to a greater tendency for cells to undergo condensation into the gel core compared to the 1-stage protocol, where collagen and alginate are crosslinked simultaneously (Wong, Tsang et al. 2019). This difference is attributed to the continuous contraction of collagen in the 2-stage process.”

Comments 7: please discuss more the reasons for the good mechanical stability revealed by the current system compared to previous studies conducted on alginate- collagen composite beside the level of MMP2 and low concentrations of calcium and sodium.

Response to the comments 7: We appreciate your feedback and would like to address your concerns regarding the discussion on the mechanical stability of the CAC ECT gel in our manuscript. We believe that the manuscript already provides a thorough discussion on this topic, emphasizing the importance of mechanical stability in relation to the immunoprotective role of the ECT device and the viability of encapsulated cells.

As mentioned in the manuscript, the mechanical stability of the scaffold is crucial for resisting the shear forces encountered during surgical procedures and the dynamic changes in the microenvironment following implantation. The integrity of the CAC scaffold, as observed by the absence of gel breakage and intact morphology in all retrieved gels, supports its ability to withstand these forces, aligning with previous studies that have reported similar findings. Furthermore, we discussed the resistance of the CAC gel to undesirable degradation, highlighting that no visible change in gel diameter was observed over a two-week period. This resistance is likely due to the relatively low concentrations of Na+ and Ca2+ in ocular tissues and potentially low levels of MMP2, which are more prevalent in elderly individuals or certain pathological conditions. These points collectively demonstrate that our CAC scaffold maintains good mechanical stability and resistance to degradation, which are essential for the long-term success of the ECT device.

However, we also include some discussion on the mechanical stability of CAC ECT gel as follows. Please refer to lines 331-360. The added information is as follows.

“There are various methods available for collagen crosslinking, and each method can significantly affect the mechanical stability of the hydrogel. Meanwhile, the crosslinking method is crucial for alginate as well. Considering the requirement for a cell-friendly environment with relatively low toxicity in cell encapsulation, we chose to crosslink collagen under mild temperature conditions, followed by alginate gelation with 0.1M CaCl2. According to our previous research results, this crosslinking method not only ensures the mechanical stability of the material but also provides optimal support for CAC ECT performance (Wong, Tsang et al. 2019).”

Comments 8: Discuss more the reasons for the selective permeability showed by the system.

Response to the comments 8: Thank you for your suggestion. We have added the appropriate discussion regarding the selective permeability of the system, which can be found in lines 421-436. The added information is as follows.

The selective permeability of the CAC ECT system can be explained through several aspects. Firstly, SEM examination of the gel surface revealed the presence of a porous structure, which serves as the physical basis for selective permeability. This structure allows the free diffusion of oxygen, nutrients, and therapeutic molecules such as GDNF, while preventing larger immune cells from entering the gel matrix. Additionally, the detection of accumulated GDNF levels in the rabbit vitreous indicates that the CAC gel effectively controls the release of GDNF while preventing the intrusion of harmful external factors, thereby ensuring therapeutic efficacy.  Studies have shown that the number of gels implanted into the vitreous does not significantly affect the viability of the encapsulated cells, suggesting that the gel's permeability is sufficient to maintain cell survival and function. Moreover, the level of GDNF secretion is closely associated with the number of implants, further confirming the effectiveness of the gel's permeability. Lastly, the selective permeability of the CAC gel plays a crucial role in immune protection by restricting the entry of immune cells, thereby preventing the host immune system from attacking the encapsulated cells and decreasing ECT device failure.

Reference:

Wong, F. S. Y., K. K. Tsang, A. M. W. Chu, B. P. Chan, K. M. Yao and A. C. Y. Lo (2019). "Injectable cell-encapsulating composite alginate-collagen platform with inducible termination switch for safer ocular drug delivery." Biomaterials 201: 53-67.

Round 2

Reviewer 1 Report

Comments and Suggestions for Authors

The authors have adequately addressed the comments raised. Thanks.